# The molecular interaction pattern of lenvatinib enables inhibition of wild-type or kinase-mutated FGFR2-driven cholangiocarcinoma

Stephan Spahn [1,13] ✉, Fabian Kleinhenz[1,13], Ekaterina Shevchenko [2,3], Aaron Stahl [4], Yvonne Rasen[1], Christine Geisler[1], Kristina Ruhm[5], Marion Klaumuenzer[6], Thales Kronenberger [2,3], Stefan A. Laufer [2,3,7], Holly Sundberg-Malek[5], Khac Cuong Bui[1], Marius Horger [8], Saskia Biskup[6], Klaus Schulze-Osthoff [7,9,10], Markus Templin[4], Nisar P. Malek[1,5,7,10,11], Antti Poso [2,3,7,12] & Michael Bitzer [1,5,7,11] ✉

Fibroblast growth factor receptor (FGFR)−2 can be inhibited by FGFR-selective or non-selective tyrosine kinase inhibitors (TKIs). Selective TKIs are approved for cholangiocarcinoma (CCA) with FGFR2 fusions; however, their application is limited by a characteristic pattern of adverse events or evocation of kinase domain mutations. A comprehensive characterization of a patient cohort treated with the non-selective TKI lenvatinib reveals promising efficacy in FGFR2-driven CCA. In a bed-to-bench approach, we investigate FGFR2 fusion proteins bearing critical tumor-relevant point mutations. These mutations confer growth advantage of tumor cells and increased resistance to selective TKIs but remain intriguingly sensitive to lenvatinib. In line with clinical observations, in-silico analyses reveal a more favorable interaction pattern of lenvatinib with FGFR2, including an increased flexibility and ligand efficacy, compared to FGFR-selective TKIs. Finally, the treatment of a patient with progressive disease and a newly developed kinase mutation during therapy with a selective inhibitor results in a striking response to lenvatinib. Our in vitro, in silico, and clinical data suggest that lenvatinib is a promising treatment option for FGFR2-driven CCA, especially when insurmountable adverse reactions of selective TKIs or acquired kinase mutations occur.

Cholangiocarcinoma (CCA) is a rare yet highly aggressive and deadly cancer with rising worldwide incidence and mortality[1]. CCA comprises a heterogeneous group of tumors with different histological and molecular subtypes[2,3]. For over a decade, a combination of gemcitabine and cisplatin has been the recommended first-line systemic treatment for advanced disease stages[4]. However, during the last few years, CCAs have become an attractive candidate for personalized

medicine approaches due to the discovery that they harbor several druggable molecular targets[5–10].

One such target is the fibroblast growth factor receptor (FGFR) 2 signaling pathway. Up to 16% of intrahepatic cholangiocarcinomas (iCCAs) harbor an *FGFR2* gene fusion that induces constitutive receptor dimerization and ligand-independent pathway activation[5,11–15]. Three drugs, pemigatinib, infigratinib, and futibatinib have been

recently approved by the FDA for previously treated iCCA tumors with FGFR2 fusions or rearrangements[16–18]. In the pivotal phase II trials that led to the approval of these drugs, independent reviews found objective response rates between 23 and 42% of all treated patients[19–21]. In addition to CCAs addicted to *FGFR2* fusion genes, activating mutations and in-frame deletions in *FGFR2* define a further group of treatment-sensitive CCAs[18,22–24]. However, these latter alterations are not yet included in the approved labels for treatment.

FGFR-targeting tyrosine kinase inhibitors (TKI) can be classified into first-generation non-selective multikinase- and second-generation selective FGFR inhibitors[25]. Most non-selective inhibitors were initially designed for other kinases but proved to harbor potent inhibitory activity towards FGFRs[26], such as ponatinib, pazopanib, nintedanib, or lenvatinib[25,27–29]. Subsequently, second-generation TKIs were developed to increase anti-FGFR activity and reduce the well-known off-target effects of multikinase TKIs[22,26]. Besides pemigatinib, futibatinib and infigratinib, further compounds are under clinical investigation for CCA, for example, erdafitinib or derazantinib[30,31]. However, by introducing these new drugs, a substantial fraction of patients develop a unique spectrum of clinically significant adverse events due to FGFR targeting. The most remarkable events are hyperphosphatemia, ocular toxicities ranging from dry eyes to severe retinal damage, and dermatologic toxicities with stomatitis, onycholysis, nail bed infections, alopecia, or calcinosis cutis[32,33]. Furthermore, several reports describe the development of acquired FGFR2 kinase domain resistances during the treatment with selective inhibitors due to multiple recurrent and polyclonal point mutations[34–37].

Much work currently focuses on the further improvement of FGFR-targeting drugs. Despite the recent development in the field of second-generation FGFR-specific inhibitors, we and others observed several profound treatment responses in FGFR2-driven CCA with non-selective TKIs, even with reduced treatment doses compared to different tumor entities[24,28,31]. In a comprehensively characterized patient cohort with CCA, we found a promising efficacy for lenvatinib in FGFR2-driven CCA. In a bedside-to-bench approach, we compared the effect of first- and second-generation FGFR-inhibiting drugs on tumor cells with patient-derived *FGFR2* alterations, including resistance-mediating point mutations. Cellular reaction patterns, proteomic and in-silico analysis demonstrate a superior activity of lenvatinib even in the presence of resistance-mediating *FGFR2* mutations. As a proof-of-principle, lenvatinib led to a long-lasting partial response in a patient with CCA who developed a kinase mutation and progressive disease during treatment with pemigatinib.

## Results

### Clinical responses of FGFR2-driven iCCA to the non-selective TKI lenvatinib

Before the approval of pemigatinib by the European Commission in 03/2021, seven iCCA patients with FGFR2 alterations were treated with the multi tyrosine kinase inhibitor lenvatinib according to a recommendation of the Molecular Tumor Board (MTB) at Tuebingen University. Notably, due to a lack of approval or fitting clinical studies, these patients could not receive a selective, second-generation FGFR-inhibiting TKI. The *FGFR2* alterations of these heavily pretreated patients are shown in Fig. 1a. One patient (370_371delinsCys & Del) of this cohort has been reported in detail previously[24]. Of note, lenvatinib led to a partial response (PR) in four of the seven patients (Fig. 1a). Two representative [[18]F]fluorodeoxyglucose (FDG)−positron emission tomography (PET) scans prior to and eight weeks after the start of lenvatinib treatment documented apparent metabolic responses (Fig. 1b). Median progression-free survival (mPFS) in this small cohort was 7.0 months, nearly three times as long as the mPFS of 2.5 months for these patients' first-line therapies (Suppl. Figure 1). Interestingly, the PFS during treatment with the established gemcitabine/cisplatin (Gem/Cis) therapy in any prior line of treatment was significantly lower

than the treatment with lenvatinib after Gem/Cis (7 vs. 2.1 months, $p \le 0.001$) (Fig. 1c). In this context, the Von Hoff model uses patients as their own control by comparing the PFS of a selected treatment with PFS values from previous lines of therapy[38]. A ratio of PFS from the investigated drug to PFS of a previous treatment of >1.3−1.5 is thereby regarded as clinically meaningful[39–41]. Of note, the PFS ratio was favorable compared to both previous and first-line therapies in 6 of 7 patients (Fig. 1a). Together with the observation of a median overall survival (OS) of more than 12 months in this heavily pretreated cohort since the start of lenvatinib therapy (Fig. 1c), these data suggest a clinically meaningful response to the treatment with lenvatinib in FGFR2-driven iCCA, even with daily doses of 12 mg or less (Fig. 1a, b).

### Therapy responses for an iCCA with a *FGFR2-AHCYL2* fusion to different TKIs

A so far unknown *FGFR2* fusion, *FGFR2-AHCYL2*, was detected in the tumor of a young female patient with an exceptional clinical course (Fig. 1d). The patient was treated with several different FGFR-inhibiting drugs. In brief, after progression on Gem/Cis and identification of the fusion, she was treated with the non-selective TKI ponatinib based on a case report[28]. However, progressive disease (PD) was already detected after 45 days. Subsequent chemotherapy was not tolerated, and lenvatinib was started as a second option to inhibit FGFR2 signaling. Intriguingly, MRI scanning revealed a partial response with normalization of initially elevated levels of the tumor marker CA19-9 six weeks later. The treatment continued until PD occurred after 9 months. A sequential liver biopsy of the progressive lesion did not find any *FGFR2* mutations as a potential explanation for tumor progression.

After an unsuccessful further treatment approach with chemotherapy and another non-selective TKI, the patient was subsequently included in a then available clinical study with infigratinib, which again led to a partial response (Fig. 1d). Of note, this observation shows that despite progression under a previous FGFR-inhibiting drug, the tumor was still addicted to FGFR signaling. However, prolonged therapy interruptions due to recurrent cholangitis led to the patient's formal study exclusion without tumor progression. A further liver biopsy was performed, which again showed the *FGFR2-AHCYL2* fusion. However, no further responses could be achieved afterward with either erdafitinib (5[th] line TKI; PFS: 0.8 months) or pemigatinib (6[th] line TKI; PFS: 0.8 months). In the meantime, further molecular diagnostics of the liver biopsy revealed the previously described *FGFR2* resistance mutation p.N549H, and a liquid biopsy additionally detected several further resistance sub- and polyclonal mutations, including a p.V564F gatekeeper and p.E565A molecular brake mutation (Fig. 1d). The patient passed away 31 months after the initiation of the treatment with lenvatinib. Taken together, this patient history demonstrates that FGFR-addicted iCCAs can show a prolonged time window for FGFR-targeted drugs; but not all selected TKIs with a preclinically known FGFR-inhibitory function could achieve a clinical response.

### Bedside to bench: generation and characterization of *FGFR2* fusion-expressing cell lines

Stable transfection of NIH3T3 cells has been used previously in several studies to investigate the transforming potential of FGFR2-fusion proteins or in-frame deletions and their sensitivity to FGFR-inhibiting drugs[11,13,18]. To investigate the effects of the newly discovered *FGFR2* fusion, we generated NIH3T3 cells to stably express the patient-specific *FGFR2-AHCYL2* fusion gene. In addition, we generated cell lines that stably expressed a second observed fusion gene from the patient cohort, *FGFR2-SH3GLB1*, and the most prevalently reported fusion in iCCA, *FGFR2-BICC1*. Characterization of these cell lines demonstrated that the expression of all three fusion genes induced a comparably increased proliferation (Fig. 2a) and anchorage-independent growth (Fig. 2b, c). Western blot analyses verified the stable expression of FGFR2 fusion proteins in the cell lines. Analysis of the downstream

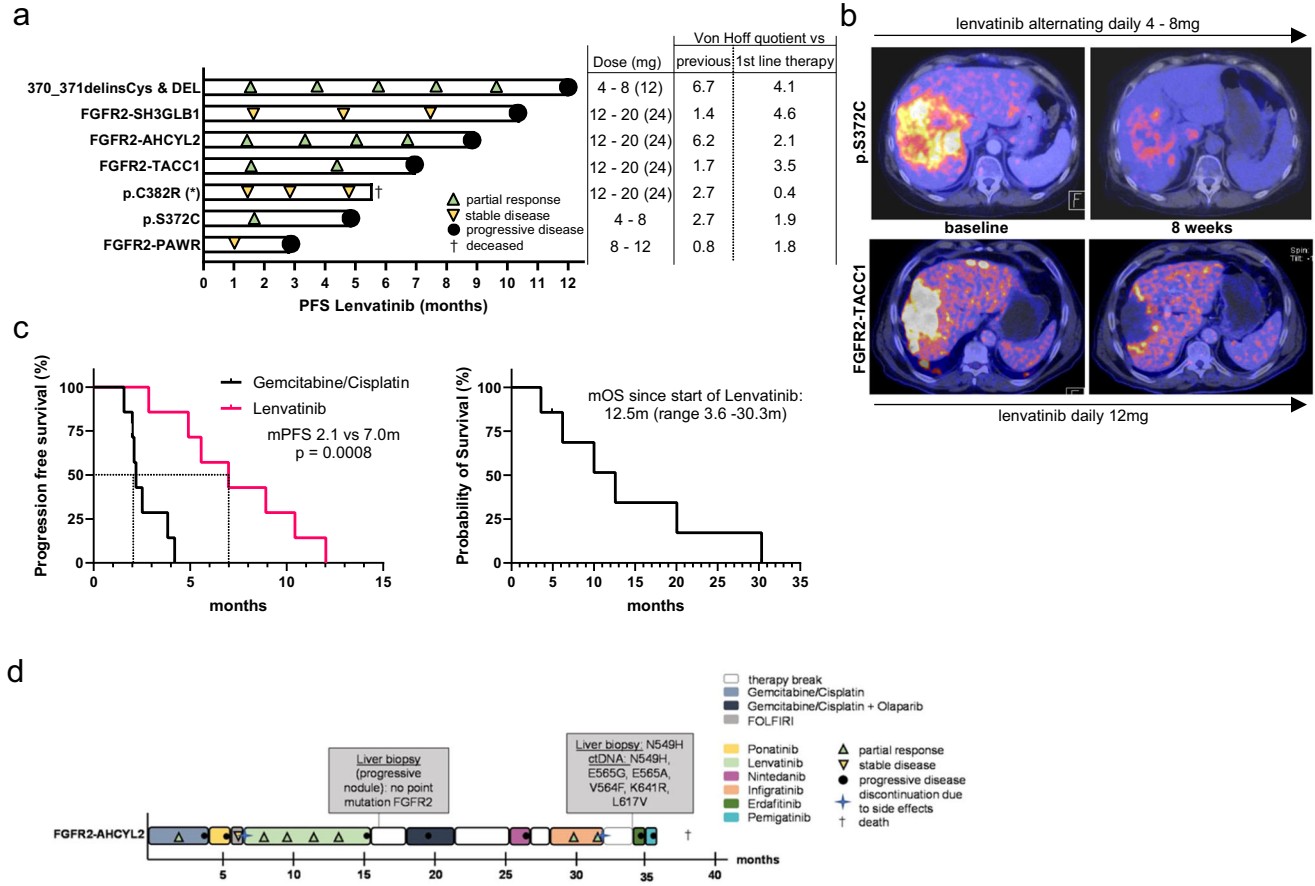

**Fig. 1 | Lenvatinib leads to clinically meaningful responses in FGFR2-driven iCCA. a** Swimmer plots illustrating the duration of individual therapy responses after the start of lenvatinib treatment. Detected molecular alterations from tumor biopsies, the applied lenvatinib dose that was used most of the treatment course with the maximal dose in brackets, and the Von Hoff quotient of lenvatinib vs. 1st line and previous therapies are shown. First-line therapies were Gem/Cis (Gemcitabine/Cisplatin) in all patients but one, who was treated with FOLFIRINOX (*). The first patient (370_371delinsCys & Del) has been reported in detail previously[24]. **b** PET-CT scans after 8 weeks of lenvatinib treatment in two patients harboring a *FGFR2* p.S372C point mutation and a *FGFR2-TACC1* fusion. The lenvatinib dose during the treatment period is shown for each patient. **c** Kaplan−Meier-survival curves demonstrating PFS of lenvatinib compared to prior Gem/Cis therapy (left panel) and the OS since the start of lenvatinib treatment (right panel). **d** Individual treatment course of a young female patient with a *FGFR2-AHCYL2* fusion undergoing treatment with different FGFR-specific and multi-targeted TKI compounds. Time points and results of *FGFR2* sequencing of liver and liquid biopsies are included. Infigratinib therapy had to be discontinued according to the clinical study protocol despite treatment response due to relapsing episodes of cholangitis. Progression was documented 43 days after end of that treatment. Two further FGFR inhibitors, erdafitinib and pemigatinib, did not reach a further response. Retrospective analysis of a liver biopsy and circulating tumor DNA (ctDNA) prior to erdafitinib revealed multiple resistance-associated point mutations.

pathways revealed that the *FGFR2* fusions predominantly led to phosphorylation and, thereby, activation of FGFR substrate 2 (FRS2), the kinases p44/42 ERK1/2, and transcription factor STAT3 (Fig. 2e). Phosphorylation of pFGF, ERK1/2 and STAT3 could be reversed through treatment inhibition with infigratinib, confirming the specificity of the phospho-specific-antibodies (Supplementary Fig. 2) Overall, these results suggest a transformation potential of the so far unknown *FGFR2-AHCYL2* fusion, comparable to the two previously described fusions, thereby qualifying these cell lines as in-vitro models for further mechanistic studies.

**Characterization of drugs with different TKI activity profiles in cells with *FGFR2*-fusions**

A direct comparison of specific and multitargeted TKIs with FGFR2-inhibitory activity in the presence of different patient-derived *FGFR2* fusion genes has not been reported in detail yet. Besides lenvatinib (targeting: VEGFR1-3, FGFR1-4, PDGFRA, KIT and RET), we selected ponatinib (targeting: ABL, PDGFRA, VEGFR2, FGFR1-2, SRC) and nintedanib (targeting: VEGFR1-3, FGFR1-3, PDGFRA/B) as multitargeted

TKIs, infigratinib (targeting FGFR1-3) and futibatinib (targeting: FGFR1-4) as FGFR-selective TKIs and cabozantinib (targeting: VEGFR2, MET, RET, KIT, FLT1,3,4, TIE2, AXL) as a multitargeted TKI without relevant FGFR-inhibitory activity as a negative control. Of note, all FGFR2-inhibiting drugs reduced cell growth in the cell line over-expressing *FGFR2-AHCYL2* (Fig. 3a). In contrast, cabozantinib, a drug without FGFR-inhibitory activity, reduced cellular proliferation stronger in the empty vector control cells than in cells transfected with *FGFR2-AHCYL2* (Fig. 3a). Treatment of the *FGFR2-SH3GLB1* and *FGFR2-BICC1* expressing cells led to similar results (Fig. 3b). Expression of *FGFR2-SH3GLB1* and *FGFR2-BICC1* led to a strong sensitization of NIH3T3 cells to the selective FGFR inhibitors futibatinib and infigratinib, resulting in an apparent strong reduction of the $IC_{50}$ values (1.8% and 7.3% compared to control transfected cells). A similar in vitro efficiency was found for the non-selective FGFR2 inhibitors lenvatinib and ponatinib, which reduced the $IC_{50}$ values of both drugs to 18.7% and 16.9%, respectively, compared to control cells. Nintedanib, which was ineffective in the above-described patient history (Fig. 1e), induced only weak responses in cells with *BICC1* and *AHCYL2*

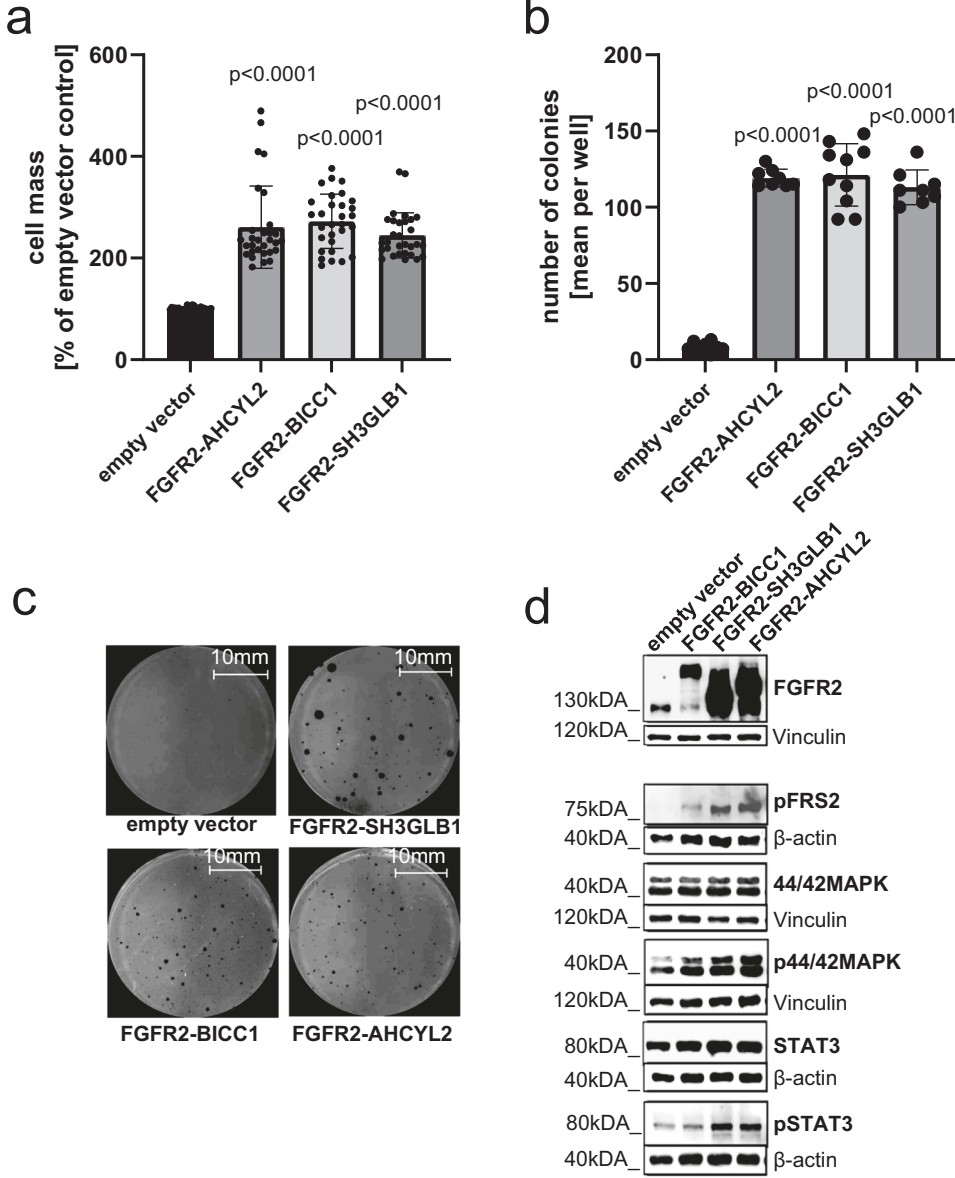

**Fig. 2 | Malignant transformation by patient-specific FGFR2-fusion proteins.**
**a** Proliferation analyses of FGFR2-fusion expressing cell lines using SRB assays after 7 days in culture, ****$P \leq 0.0001$ compared to control transduced NIH3T3 cells with one-way ANOVA with Dunnett's multiple comparison test. Bars represent mean ± SD ($n = 30$ vials examined over 10 independent experiments). **b** Quantification of soft agar colony formation after 21 days with *FGFR2*-fusion-expressing cell lines and a control cell line transfected with the empty vector, ****$P \leq 0.0001$ compared to control transfected NIH3T3 cells with one-way ANOVA with Dunetts's multiple

comparison test. Bars represent mean ± SD ($n = 8$ vials (empty vector, FGFR2-AHCYL2, FGFR2-SH3GLB1), $n = 10$ vials (FGFR2-BICC)) examined over 3 independent experiments. **c** Representative images of soft agar assays with the indicated cell lines after 21 days. **d** Representative Western blot analysis of FGFR2 downstream signals in NIH3T3 cell lines stably transfected with patient-derived *FGFR2* gene fusions (*FGFR2-BICC1*, *FGFR2-SH3GLB1*, *FGFR2-AHCYL2*) or control transfected NIH3T3 cells ($n = 3$ biologically independent samples).

fusions and no response in the *FGFR2-SH3GLB1*-expressing cell line (Fig. 3b). Besides inhibiting proliferation, colony formation assays showed comparable inhibitory activity of selective and non-selective FGFR inhibiting TKIs in *FGFR2-AHCYL2* transfected cells (Supplementary Fig. 3A, B).

So far, the clinical and in vitro results suggest that both FGFR2-selective and multi-target TKIs could be employed to inhibit the cell growth of *FGFR2*-fusion-positive tumors. To dissect the molecular mechanism of selective and non-selective TKIs for FGFR signaling in more detail, we characterized the *FGFR2-AHCYL2* expressing cell line via DigiWest, a high-throughput proteomic approach to analyze cellular signaling pathways[42] (Supplementary Fig. 4). To allow comparison, concentrations of TKIs were selected leading to

approximately 50% cell mass reduction in the SRB assays. First, we looked at phosphorylated FGFR2 (p-FGFR2) and found a similar inhibition of FGFR2 phosphorylation for all drugs (Fig. 3c). Of note, no p-FGFR2 signal was detected in control transfected NIH3T3 cells. Further study of downstream targets revealed that FGFR-selective TKIs exclusively inhibited members of the MAPK pathway, such as ERK1/2 (p- Thr202/Tyr204) and RSK1 (p90RSK, p-Thr573), SHP2 (p-Tyr542), and the downstream target c-Myc (p-)Thr58/Ser62 in *FGFR2-AHCYL2* transfected cells (Fig. 3d, Supplementary Data 1). In contrast, the non-selective TKIs additionally inhibited MAPK-unrelated proteins, such as p70S6 kinase (Thr389)) and S6 ribosomal protein (Ser235/236), which are involved in Jak/STAT or PI3K/AKT/mTOR pathways and inhibited a broader spectrum of MAPK-related

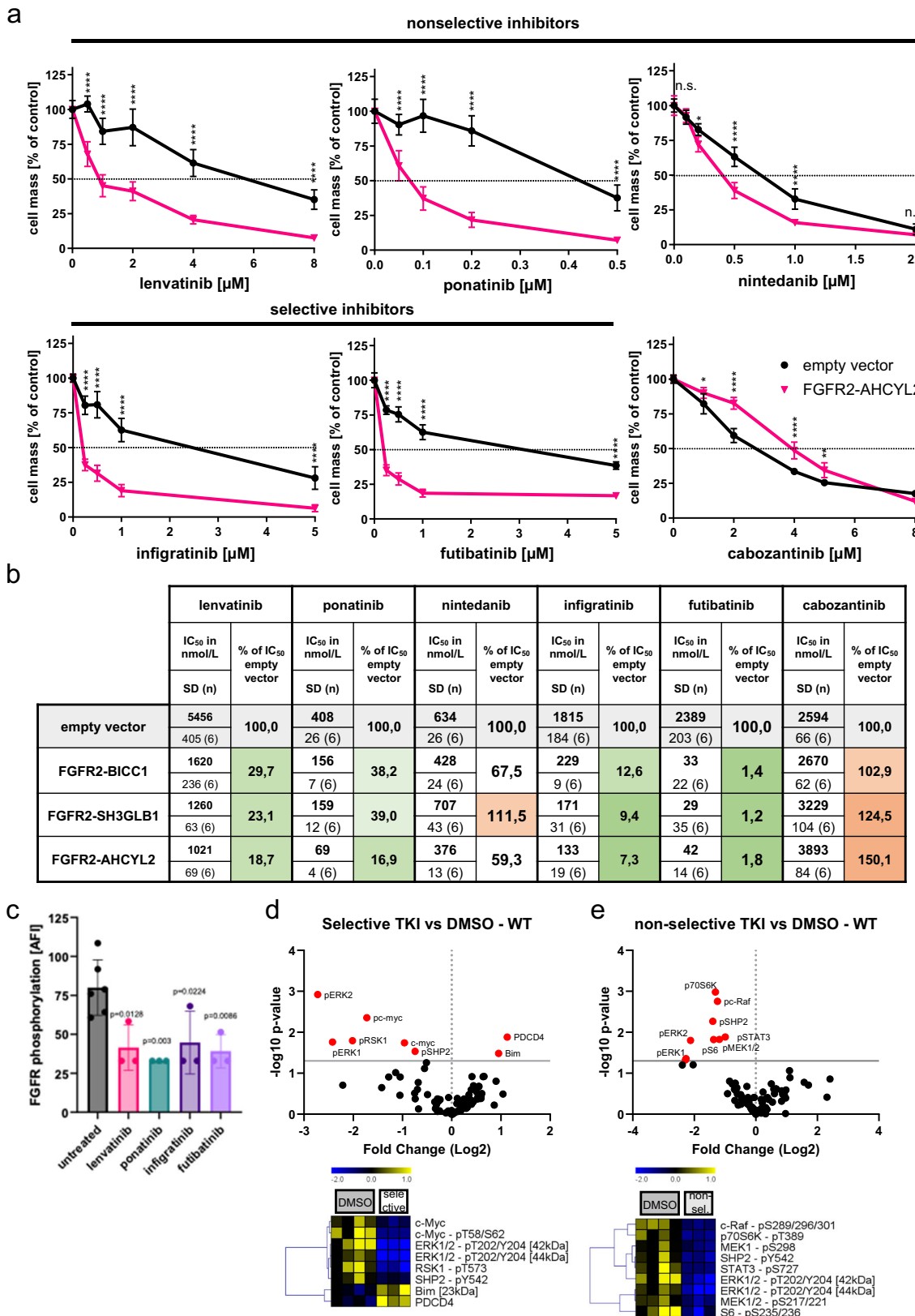

proteins, such as MEK1/2 (p-Ser217/Ser221) and c-RAF (p-Ser289/296/301) (Fig. 3e, Supplementary Fig. 5a, Supplementary Data 1). Interestingly, this broader inhibition of additional signaling pathways by the non-selective TKIs was particularly prominent in the

*FGFR2*-fusion expressing cell line, which might be important in the context of treatment responses and the acquisition of therapy resistance. No significant difference was observed in the phosphorylation status of mTOR or p38 (Supplementary Data 1).

**Fig. 3 | FGFR-selective and non-selective TKIs in FGFR2-fusion positive cells.**
**a** Proliferation analyses in NIH3T3 cells with the *FGFR2-AHCYL2* fusion employing SRB assays after 7 days of treatment. Applied drugs included the non-selective TKIs lenvatinib, ponatinib and nintedanib; the selective FGFR inhibitors infigratinib and futibatinib; and cabozantinib, a TKI without FGFR2-inhibitory activity as a negative control. *$P \leq 0.05$, **$P \leq 0.01$, ****$P \leq 0.0001$ compared to control transduced NIH3T3 cells with one-way ANOVA with Dunnett's multiple comparison test. Data are presented as mean ± SD ($n = 6$ independent experiments). **b** IC$_{50}$ values of different drugs for NIH3T3 cells expressing the indicated construct in nmol/L ± SD and the respective percentage of the value for empty vector-transfected cells. Cell background gradient: green = lower, red = higher IC$_{50}$ compared to empty vector. **c–e** DigiWest protein profiling analysis of *FGFR2-AHCYL2* fusion cell line treated with selective and non-selective TKIs. **c** Phosphorylation status (accumulated

fluorescent intensity) of FGFR2 (Y653/654) in *FGFR2-AHCYL2* samples. Bars represent mean ± SD ($n = 6$ (untreated) and $n = 3$ (TKI-treated) biologically independent samples), One-Way-ANOVA with Dunnett's multiple comparison test. Volcano plot and hierarchical cluster (HCL) analysis of proteins and phosphoproteins significantly different between (**d**) selective inhibitors (infigratinib and futibatinib, $n = 3$ biologically independent samples) and (**e**) non-selective inhibitors (lenvatinib and ponatinib, $n = 3$ biologically independent samples) compared to control (DMSO, $n = 4$ biologically independent samples) in samples from *FGFR2-AHCYL2* cells (two-sided T-test, Welch, $P \leq 0.05$). Expression values were normalized to total protein signals across all samples for a given analyte, median-centered and Log-2 transformed. Shown are selected signaling proteins, the full DigiWest data set is included in the Supplementary Data file 1. Hierarchical cluster analysis was performed using Pearson correlation and complete linkage.

## Generation and characterization of cell lines expressing resistance mutations within the *FGFR2* kinase domain

The selection of resistant subclones during the treatment with a selective FGFR2 inhibitor is a well-described mechanism and a major clinical concern[34–36,43]. The patient with the *FGFR2-AHCYL2* fusion finally developed polyclonal resistances after the treatment with infigratinib (Fig. 1d). In the preceding treatment with lenvatinib over 9 months, no resistance mutation was found. As our proteomics analysis revealed different inhibitory patterns of FGFR-specific and multitargeted TKIs, we speculated that resistance-mediating point mutations in *FGFR2* might be a predominant problem that arises during the treatment with FGFR-specific TKIs. Hence, we generated two additional cell lines that expressed either the *FGFR2-AHCYL2* construct with the previously described p.V564F "gatekeeper-mutation" (hereafter called *FGFR2-AHCYL2 plus* p.V564F) or p.E565A "brake-mutation" (*FGFR2-AHCYL2 plus* p.E565A). No differences in proliferation or expression levels of (mutated) FGFR2-fusion protein were noted between cells transfected with *FGFR2-AHCYL2 plus* p.V564 or plus p.E565A or *FGFR2-AHCYL2* without these mutations (Supplementary Fig. 6).

Employing different FGFR-inhibitory drugs, cell viability assays showed that the *FGFR2-AHCYL2 plus* p.V564F and the *FGFR2-AHCYL2 plus* p.E565A mutations, as expected, caused resistance to infigratinib and, to a lesser extent, to futibatinib (Fig. 4a). Nevertheless, IC$_{50}$ values for futibatinib were up to 28.5 higher in the cell lines expressing the *FGFR2-AHCYL2* plus point mutations compared to the cells transfected with *FGFR2-AHCYL2* (Fig. 4b). In contrast, only slight effects on IC$_{50}$ (1.9-fold change for FGFR2-*AHCYL2* plus p.V564F; 2.7-fold change for FGFR2-*AHCYL2* plus p.E565A compared to *FGFR-AHCYL2*) were noted for lenvatinib, with a clear inhibition of cell growth, especially at low concentrations. Ponatinib was still active in the presence of the p.V564F "gatekeeper-mutation" in our model, yet did not show an impact on the proliferation of cells with the *FGFR2-AHCYL2* plus p.E565A "brake-mutation" (Fig. 4a).

To further characterize the different response patterns of selective and multitargeted TKIs in the presence of resistance mutations, we performed a proteomic analysis of the *FGFR2-AHCYL2* plus p.V564F "gatekeeper mutation" harboring cell line. Interestingly, treatment with the multitargeted TKIs lenvatinib and ponatinib, as well as with the irreversible pan-FGFR inhibitor futibatinib, led to reduced phosphorylation of FGFR2 (Fig. 4c). In clear contrast, the p.V564F mutation prevented dephosphorylation of FGFR2 after infigratinib treatment (Fig. 4c). DigiWest analysis of downstream signaling pathways in infigratinib- or lenvatinib-treated cells demonstrated that lenvatinib could still inhibit the phosphorylation of non-MAPK signaling proteins, such as AKT (Ser473), mTOR (Ser2481 und Ser2448), p70S6K (Thr389), and eIF4E (Ser209), which are mainly involved in PI3K/AKT/mTOR signaling (Fig. 4d, e, Supplementary Fig. 5b, Supplementary Data 1). This was underlined by directly comparing the two treatments to each other. Lenvatinib exhibited a conserved inhibitory differential effect for

phosphorylation of FGFR2 (Tyr653/Tyr654), mTOR (Ser2481), eIF4E (Ser209) as well as total mTOR (Fig. 4f, Supplementary Data 1). This not only underscores the differential effect on FGFR2 phosphorylation (see Fig. 4c) but also highlights the persistent inhibition of downstream pathways of lenvatinib, suggesting that the sustained activity of lenvatinib is likely due to conserved direct FGFR2 inhibition even in the presence of the p.V564F mutation.

## In-silico modeling demonstrates a favorable interaction pattern of lenvatinib within the ATP-binding pocket of FGFR2

To gain more insight into the molecular interaction of lenvatinib with wild-type and mutated FGFR2, we performed in-silico modeling applying long time-scale (in total 48 μs) molecular dynamics (MD) simulations for wild-type FGFR2 and the three exemplarily selected mutations E565A, V564F, and N549K (Fig. 5a, b). Resulting trajectories were subjected to interaction analysis (Fig. 5c, Supplementary Tables 1–3), evaluation of binding free energy and its components with molecular mechanics energies combined with the generalized Born and surface area continuum solvation (MM-GBSA) (Supplementary Figs. 7–12), and TKIs study of torsional profiles (Supplementary Figs. 13–15).

First, we conducted an interaction analysis of the FGFR2 molecular brake, a regulatory element comprising a molecular triad of residues N549, E565, and K641, governing autoinhibition[44–48]. Despite previous indications of its significance in drug resistance, our investigation did not reveal notable differences in the interactions involving the molecular brake across selected TKIs (chemical structures are shown in Supplementary Figs. 13–15) and mutations (Supplementary Table 1). Hence, we observed that the molecular brake changes do not play a prominent role in our studied mutation-inhibitor combinations and omitted them from further investigation into this aspect.

Further analysis highlighted lenvatinib's prominent interaction engagement, excelling infigratinib and pemigatinib in both WT and mutant FGFR2 settings (Fig. 5c, Supplementary Table 2A and Suppl. Discussion). These observations suggest that lenvatinib possesses a dynamic ability to alter its interaction pattern in response to mutations, thus fine-tuning its binding to the evolving protein binding pocket. We extended these analyses to four further mutations that have been described in the context of therapy resistance to FGFR2-specific TKIs[21,35,44] or gain of function[44], namely N549D, V562L, V564I, and E565G (24 μs), which showed similar results (Supplementary Table 2B). This adaptability was further reflected in the prevalence of hydrophobic interactions, with lenvatinib displaying superior hydrophobic engagement compared to the other two TKIs (Supplementary Table 3). This observation was further supported by lenvatinib's leading ligand efficacy (Supplementary Fig. 7) and the MM-GBSA binding free energy components (Supplementary Figs. 8–12, Supplementary Discussion).

To unravel the basis of Lenvatinib's adaptability, we delved into the torsional profiles of the TKIs within the gate area and back cleft of

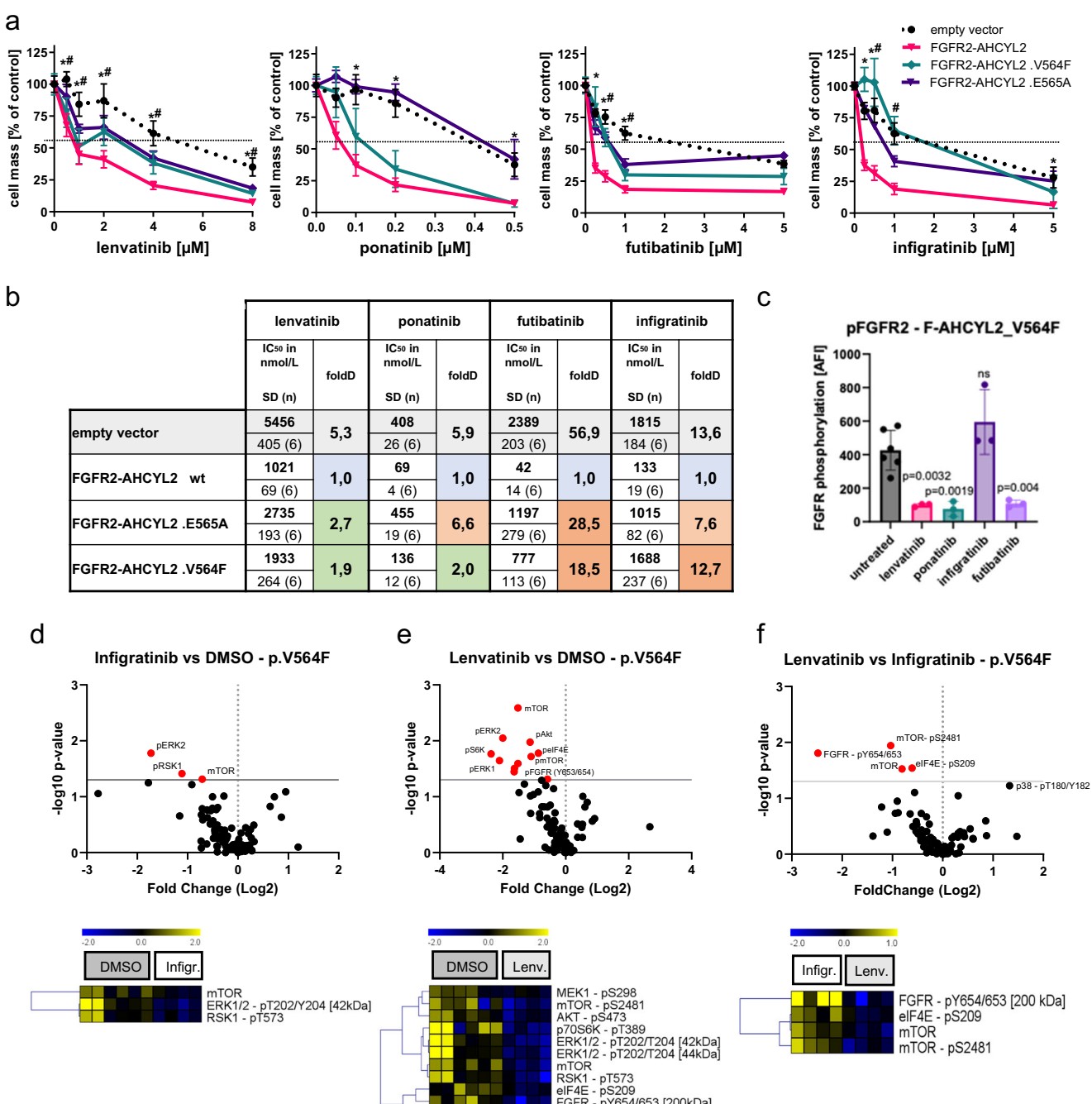

**Fig. 4 | Non-selective and selective TKIs with FGFR2-inhibitory activity in the presence of resistance-mediating mutations. a** Proliferation analyses using SRB assays in the NIH3T3 cell lines harboring *FGFR2-AHCYL2* fusions with either wild-type (WT), p.E565A, or p.V564F mutations in the kinase domain of the *FGFR2-AHCYL2* fusion. Cell lines were analyzed after 7 days of treatment with the indicated non-selective TKIs lenvatinib and ponatinib or the FGFR-selective TKIs infigratinib and futibatinib, *$P \leq 0.05$ for FGFR2-AHCYL2 plus p.E565A vs empty vector control at the indicated drug dose, #$P \leq 0.05$ for FGFR2-AHCYL2 plus p.V564F vs empty control at the indicated drug dose using two-way ANOVA with Tukey's multiple comparison test. Data are presented as mean ± SD ($n = 6$ independent experiments). **b** IC$_{50}$ values and fold difference (foldD) of different drugs for NIH3T3 cells expressing the indicated construct in nmol/L ± SD and their respective percentage of the value for empty vector-transfected cells. Cell background gradient: green = lower foldD, red = higher foldD of IC$_{50}$ compared to

FGFR2-AHCYL2 wildtype cells. **c–e** DigiWest protein profiling of NIH3T3 cells expressing the p.V564F_*FGFR2-AHCYL2* fusion gene after treatment with selective or non-selective TKIs. The full DigiWest data set is included in the Supplementary Date file 1. **c** Phosphorylation status (accumulated fluorescence intensity) of FGFR (Y653/654) in p.V564F *FGFR2-AHCYL2* samples. Bars represent mean ± SD ($n = 6$ (untreated) and $n = 3$ (TKI-treated) biologically independent samples), One-Way-ANOVA with Dunnett's multiple comparison test. Volcano Plot and hierarchical cluster (HCL) analysis of proteins and phosphoproteins that significantly differed between (**d**) infigratinib-treated ($n = 4$ biologically independent samples), (**e**) lenvatinib-treated ($n = 4$ biologically independent samples) compared to control (DMSO, $n = 6$ biologically independent samples) or (**f**) lenvatinib- vs. infigratinib-treated ($n = 4$ biologically independent samples) p.V564F *FGFR2-AHCYL2* samples (two-sided T-test, Welch, $P \leq 0.05$).

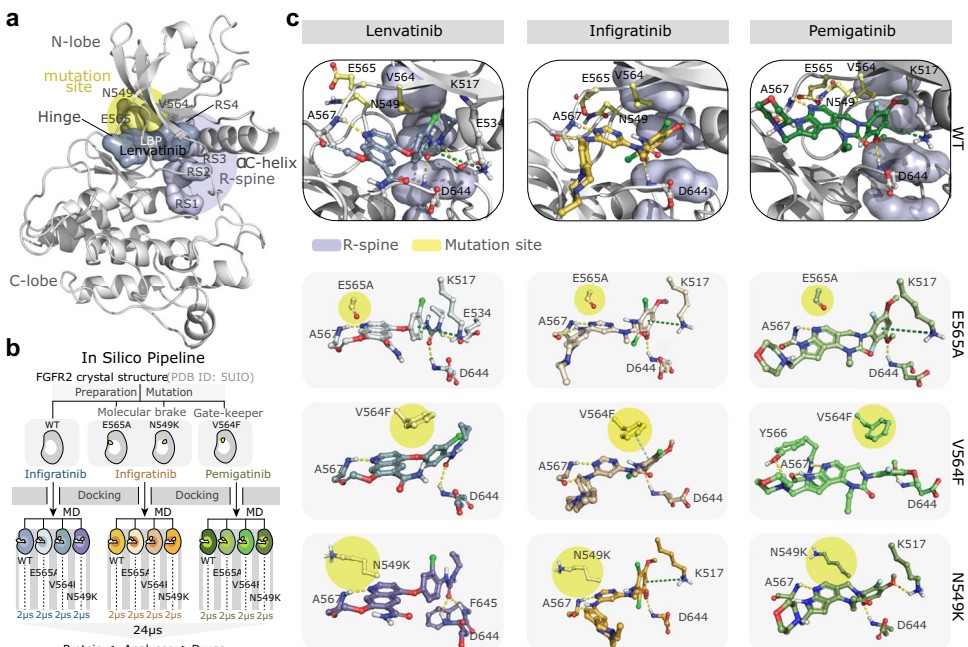

**Fig. 5 | In-silico modeling suggests that lenvatinib adapts better to FGFR2 kinase mutations than infigratinib or pemigatinib. a** Structure of wild-type FGFR2 kinase domain bound to lenvatinib in the ATP-binding cleft. The location of the mutations E565A, V564F, and N549K is shown in the yellow circle, lenvatinib is shown in blue and, the regulatory R-spine residues R1-R4 in violet; the R-spine is shown in assembled (active) conformation. **b** In-silico pipeline for analyses of the FGFR2 mutation impact on drug effectiveness. **c** Lenvatinib, infigratinib and pemigatinib interactions with wild-type FGFR2 and different mutants. The pictures show representative snapshots from Molecular Dynamics trajectories with contacts occurring in more than 20% of the simulation time (full data is available in Supplementary Table 2). Ligands are represented with stick models, colors of carbon atoms are as in **b**, and colors of the structural elements are as in **a**. Studied mutations are highlighted with yellow circles. A light blue dashed line represents π–π stacking, a green line represents π-cation, a yellow line represents H-bond, and a dark blue line represents aromatic H-bond.

FGFR2. Our analysis highlighted a dynamic flexibility of the lenvatinib terminal flexible cyclopropyl moiety in contrast to the more rigid terminal dimethoxyphenyl moiety of infigratinib and pemigatinib, which suggests that lenvatinib is sterically more flexible in the presence of FGFR2 mutations (Supplementary Figs. 13–15).

A more detailed description of the in-silico work is given in Supplementary Material, Results and Discussion. Taken together, these observations and the in vitro data reveal that lenvatinib can adapt better to FGFR2 mutations than the investigated FGFR-specific TKIs.

### Lenvatinib overcomes resistance to pemigatinib in a patient with *FGFR2-BICC* N549K resistance mutation

Intrigued by these results, we treated a female patient with lenvatinib after development of progressive disease during a previous treatment with pemigatinib. Initially, the patient had a partial response to pemigatinib but developed hyperphosphatemia and additional side effects, such as complete hair loss and recurrent nailbed inflammations. Of note, during treatment with pemigatinib, also increased hepatic calcification (Supplementary Fig. 16) appeared as a phenomenon that had been previously described in a patient treated with infigratinib[27]. Unfortunately, the patient developed progressive disease after a PFS of 13 months during pemigatinib therapy.

A further biopsy of a liver lesion was then performed that revealed a *FGFR2* N549K brake mutation. We therefore started therapy with lenvatinib. Strikingly, a follow-up CT scan taken 29 days later showed a consistent shrinkage of all liver lesions accompanied by a considerable reduction of the rim enhancement (Fig. 6). Besides mild hypertension, no phosphate elevation occurred, the nailbed inflammations disappeared, and hair growth returned. Up to date, four further follow-up scans confirmed the ongoing effective partial response, showing an even more profound reduction of tumor manifestations, with some completely disappearing (Fig. 6).

## Discussion

Protein kinase inhibitors have emerged as essential tools in the armamentarium to treat cancer, resulting in the approval of more than 70 new drugs since the first approval of imatinib in 2001[49]. Each drug has different pharmacological properties and individual kinase-inhibitory profiles that might be utilized to personalize TKI selection. Here, we describe a cohort of 7 patients with FGFR2-driven CCA that have been treated with the multikinase TKI lenvatinib as first line FGFR2 targeted therapy and one additional patient that was treated with lenvatinib after developing resistance to the treatment with the selective FGFR2 inhibitor pemigatinib.

In the 7 patients, the growth modulation index (GMI), which is based on the assumption that the time to progression tends to be shorter in each subsequent treatment line in advanced cancers[38,40,50–52], was analyzed as an intra-patient comparison of different treatment regimens. Of note, six of the seven patients treated with lenvatinib had a GMI value > 1.3 regarding the previous and even compared to the first-line therapy (Fig. 1a), suggesting lenvatinib to be efficient in this treatment setting. This observation was further supported by (i) PET-CT scans documenting metabolic responses during the first weeks of treatment and (ii) the comparison of the median PFS of 7.0 months for the treatment with lenvatinib to 2.1 months for the first-line treatment with Gem/Cis in this cohort.

One of the patients had an *FGFR2-AHCYL2* fusion, which to our knowledge has not been described so far, although *FGFR2-AHCYL1*, a structurally related fusion gene, was reported previously[13]. To characterize the fusion protein on the cellular level, we generated NIH3T3 cell lines to stably overexpress *FGFR2-AHCYL2*, *FGFR2-SH3GLB1*, or *FGFR2-BICC1*. All three fusions conferred enhanced proliferation, colony formation, or activation of the downstream targets FRS2, ERK1/2, and STAT3.

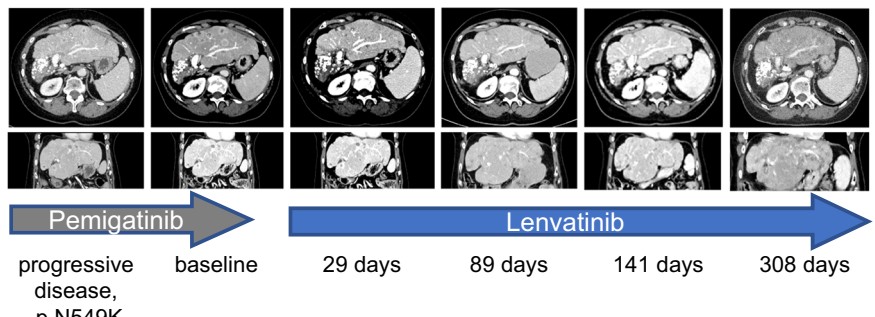

**Fig. 6 | Lenvatinib leads to a partial response after progression to pemigatinib in the presence of the N549K resistance-mediating mutation.** Axial contrast-enhanced CT of the liver in the portal-venous phase after progressive disease to pemigatinib in the presence of an N549K kinase mutation detected in a liver biopsy of a progressive lesion. From left to right: progressive disease during therapy with pemigatinib; baseline CT scan prior to lenvatinib; first follow-up scan 29 days later with consistent shrinkage of all liver lesions, accompanied by considerable reductions of the rim enhancement; second follow-up 89 days after the start of lenvatinib, showing a confirmation of the treatment responses with even more efficient reduction of tumor manifestations, some of them having completely disappeared; further confirmation of the ongoing response at days 141 and 308.

Interestingly, the clinical case with *FGFR2-AHCYL2* showed treatment responses to lenvatinib and infigratinib, but not to ponatinib or nintedanib (Fig. 1d). A comparison of the $IC_{50}$ values of different FGFR-inhibitory drugs revealed an increased sensitivity for lenvatinib, ponatinib, infigratinib, and futibatinib in cells transfected with the fusion gene compared to control cells. In contrast, nintedanib showed only minor effects in fusion-positive cells. Whereas the cellular response to nintedanib might mirror the missing response during our patient's treatment, it cannot explain the absent response to ponatinib. Of note, this case illustrates several important features of FGFR2-addicted CCAs. First, FGFR2 can remain a relevant drug target despite progression during treatment. It is tempting to speculate that at the end of the patient's treatment, a FGFR inhibitor targeting most of the polyclonal resistance mutations would have led to a further clinical benefit. Second, no kinase mutation was detected after progression despite the treatment with lenvatinib for more than nine months. For lenvatinib, feedback activation of the EGFR-PAK2-ERK5 signaling axis has been described as a resistance mechanism in hepatocellular carcinoma[53]. Thus, other factors than kinase mutations might have to be considered after the progression of FGFR2-driven CCAs during therapy with lenvatinib.

One of our initial hypotheses was that the influence of resistance-mediating kinase mutations might be different between FGFR-selective and multikinase TKIs. To investigate this possibility, we selected two previously described point mutations in *FGFR2*: p.V564F and p.E565A. The valine residue in the drug-binding pocket of the kinase domain is conserved in FGFR1-4[25]. Amino acid substitutions at this gatekeeper position alter the mode of drug-FGFR interactions and are called gatekeeper mutations[22]. An autoinhibitory brake that is made up of three main residues, an asparagine (N), a glutamate (E), and a lysine (K), is called the NEK or molecular brake triad. Within FGFR2, this conformation is located at p.N549, p.E565, and p.K641[25]. As molecular brake residues mediate the autoinhibition of FGFR kinases, mutations in this region lead to constitutive kinase activation[22].

Cell viability assays surprisingly revealed that despite the presence of the p.V564F gatekeeper or the p.E565A mutations, lenvatinib significantly inhibited cell growth, even at low concentrations. Moreover, lenvatinib conferred inhibition of downstream targets in protein profiling analyses, besides typical FGFR2 downstream signaling pathways, which might be important in the context of treatment responses or acquisition of resistance and might be a beneficial additive consequence of the more unselective nature of lenvatinib. To gain further insights into mutation impact on drug binding, we applied an in-silico pipeline to model lenvatinib-, infigratinib-, and pemigatinib-FGFR2 bound systems, encompassing WT, N549K, N549D, E565A, E565G, V562L, V564F, and V564I mutations, followed by classical all-atom MD simulations. FGFR2 MD studies were previously reported in work by Sangeetha et al.[54]; however, with a total timescale of 48 μs in our work, we go far beyond the reported data. Our results suggest that lenvatinib's superior inhibitory performance occurs not only due to a more favorable interaction pattern but to a set of additional factors such as hydrophobic stabilization, increased flexibility of the lenvatinib in the gate area and back cleft and superior free energy ligand efficiency (for additional information see Supplementary Methods, Results and Discussion).

As an outcome of our work, we treated a patient after progression to the selective inhibitor pemigatinib and the development of a resistance mutation with lenvatinib, which revealed an impressive and, so far, durable response. Furthermore, as an additional clinically relevant observation, the class-specific side effects of pemigatinib, which limited the quality of life in that patient, subsided after the end of pemigatinib treatment and did not return during the treatment with lenvatinib. In contrast to the selective FGFR inhibitors infigratinib and pemigatinib, lenvatinib has a higher inhibition efficiency of FGFR2 than FGFR1 (see Supplementary Table 4). We hypothesize that this could be one reason for the different spectrum of adverse effects.

Our results stimulate hypotheses for exploratory studies that could guide the optimal inclusion of lenvatinib in the treatment algorithm of FGFR2-driven CCA, such as to compare the appearance of kinase resistance mutations during the treatment with lenvatinib or specific-FGFR inhibiting TKIs by repeated liquid biopsies or therapy responses despite the presence of resistance-mediating mutations.

In conclusion, we demonstrate the potential of the unspecific TKI lenvatinib, even at low doses, to treat CCA addicted to FGFR2 signaling even in the presence of resistance mutations. Our observations have several clinical implications. First, in the case of insurmountable characteristic adverse reactions of FGFR-specific TKIs, lenvatinib seems to be an efficient alternative. Second, our data suggest that due to its broader activity on intracellular signaling events and increased flexibility in the kinase pocket, lenvatinib can overcome and might prevent or delay the development of resistance-mediating FGFR2 mutations.

## Methods

### Patients

All presented patients were referred to the Molecular Tumor Board (MTB) at the University Hospital Tuebingen. The translational study was reviewed and approved by the local ethics committee of the medical faculty (714/2019BO2). Off-label treatments were recommended by the MTB, which consists of an interdisciplinary team including experts in clinical and translational oncology, pathology, bioinformatics, molecular biology, radiology, and human genetics[6]. All

patients gave written informed consent before treatment with lenvatinib. Before genetic tumor analysis, patients were consulted by a specialist in clinical genetics. Tumor genetic analysis [liquid biopsy, next generation sequencing (NGS), transcriptome or whole-exome sequencing (WES)] were performed by CeGaT GmbH, Tuebingen, the Institute of Medical Genetics and Applied Genomics, Tuebingen, or inside the Molecularly Aided Stratification for Tumor Eradication Research (MASTER) precision oncology program at the National Center for Tumor Diseases/German Cancer Consortium (NCT/DKTK), as previously described[6,55]. To assess treatment efficacy, CT or MRI scans were reviewed for complete (CR) and partial response (PR), stable disease (SD) or progressive disease (PD) by an experienced radiologist based on principles of RECIST version 1.1. criteria. Staging examinations were performed every 4–12 weeks.

## Cell culture and chemicals
The NIH3T3 cell line was a kind gift by Wolfgang Neubert (Max Planck Institute for Biochemistry, Martinsried, Germany) and authenticated by ATCC using Short Tandem Repeats (STR). Cells were grown in a humidified atmosphere at 37 °C in 5% $CO_2$ in Dulbecco's Modified Eagle's Medium (DMEM)−high glucose (Sigma-Aldrich, Taufkirchen, Germany) complemented with 10% FBS and 1% penicillin-streptomycin (Thermo Fisher Scientific, Schwerte, Germany) and routinely tested for mycoplasma with a DAPI test. Futibatinib was purchased from Cayman Chemicals (Ann Arbor, MI, USA); all other TKIs and gemcitabine were purchased from Selleckchem (Houston, TX, USA).

## Cloning strategy and stable transfection
The *FGFR2-AHCYL2*, *FGFR2-SH3GLB1*, and *FGFR2-BICC1* fusion genes were generated and cloned into the pcDNA3.1(+)P2A-eGFP vector by GenScript (New Jersey, U.S). GenScript used site-directed mutagenesis to introduce the p.V564F and p.E565A SNVs into the *FGFR2-AHCYL2* fusion. NIH3T3 cells were stably transfected with 2 µg of the linearized vectors using Effectene® transfection reagent (Qiagen, Hilden, Germany). Single clones were selected using Geneticin (G418 disulfate salt; Biochrom, Berlin, Germany). Empty pcDNA3.1(+)P2A-eGFP transfected NIH3T3 were used as control.

## Western blot analysis
Primary antibodies are specified in Supplementary Table 5. To determine the relative protein abundance, densitometry of the total surface area of the respective bands was performed and normalized to the respective band of β-actin or vinculin using ImageJ. Preparation of cells and technical details are shown in Supplementary Methods.

## Cell viability assay
For proliferation and $IC_{50}$ measurements using different compounds, sulforhodamine B-assays (SRB) were performed as described in Supplementary Methods.

## Soft agar colony formation assays
2000 cells were suspended in DMEM with 20% FBS and 0.35% Difco™ Noble Agar (BD Biosciences, Heidelberg, Germany) and the indicated substances. Subsequently, the cells were seeded in six-well plates plated with DMEM with 20% FBS with 0.7% Difco™ Noble Agar (BD Biosciences). After 21 days in culture, colonies were stained overnight using Iodonitrotetrazolium chloride (violet) (Sigma-Aldrich). Colonies were counted using Image J.

## DigiWest multiplex protein analysis
DigiWest was performed as described previously[42] using 10–12 µg of cell lysate per sample. A detailed description of the DigiWest procedure is included in the Supplementary Methods. A scheme outlining the DigiWest workflow can be found in Supplementary Fig. 4, the employed antibodies are described in the Supplementary Data 1and

the complete data set of all DigiWest investigations is included in the Supplementary Data 1. Signal quantification and analysis were performed using an Excel-based analysis tool. MEV 4.9.0 was used for heatmap generation and respective statistics (Welch´s T-Test, group comparison). For DigiWest Supplementary Figs. 4A/B, after correction for experimental variation, groups were compared to DMSO controls using Welch´s ANOVA with Dunnett´s Multiple Comparisons Test.

## In-silico modeling
**Preparation of FGFR2 WT, N549K, N549D, E565A, E565G, V562L, V564F, and V564I systems.** At the time of the analysis 47 FGFR2 structures were available in the RCSB Protein Data Bank with 31 containing the kinase domain (https://www.rcsb.org, accessed 13/07/2022). The crystal structure used for the modeling was FGFR2 harboring an E565A/K659M double mutation (PDB ID: 5UIO[56]) with the resolution of 2.05 Å, comprising 324 amino acids. As E565A was of interest, we reversed the K659M mutation using Maestro (2021.3) with further hydrogen bond assignment and energy minimization with Protein Preparation Wizard[57] (Maestro 2021.3, Schrödinger LLC, New York, NY, USA). The rotamer position of the reversed K659 residue was checked from the wild-type FGFR2 crystal structure (PDB ID: 2PVF[44]). To maintain the consistency in system preparation and annihilate potential artificial errors in further system comparisons, we subsequently reversed the E565A mutation. In the obtained wild-type FGFR2 we introduced separately N549K, N549D, E565A, E565G, V562L, V564F, and V564I mutations, followed by hydrogen bonds and energy minimization with the same protocol as above. The rotamer position of gatekeeper V564F mutation was comparable to those in the corresponding crystal (PDB ID: 7KIA[58]), the FGFR2 N549K, N549D, V564I crystal structure has not been solved to date.

**Docking of lenvatinib, infigratinib and pemigatinib in prepared FGFR2 systems.** To generate the grid for further docking, we aligned the FGFR1−lenvatinib crystal structure (PDB ID: 5ZV2[59]) with our newly generated WT model. FGFR1 and FGFR2 share 87% sequence similarity in the kinase domain, which refers to comparable ligand position inside the ligand-binding pocket. After superimposing, the FGFR1 structure was deleted and the remaining lenvatinib in the binding pocket was used for SiteMap[60,61] binding site evaluation. 5 Å buffer distance from lenvatinib with more restrictive definition of hydrophobicity was used for evaluation. The output from SiteMap was used for the receptor grid generation with Glide[62–64].

Lenvatinib, infigratinib and pemigatinib were prepared using LigPrep (Schrödinger Release 2021-3: LigPrep, Schrödinger, LLC, New York, NY, 2021; default settings) to generate the 3D conformation of the compounds, their ionization states to pH 7.0 ± 1.0, and calculate their charges. Subsequently, prepared ligands were docked into the FGFR2 WT, N549K, N549D, E565A, E565G, V562L, V564F, and V564I model using Glide[62,64] (default settings, XP-accuracy). To validate model precision, post-docking validation of infigratinib pose was made by comparison to the co-crystallized FGFR1−infigratinib complex (PDB ID: 3TTO[65]); crystal structures encompassing pemigatinib have not been solved to date.

Detailed information on Molecular Dynamics Simulations, interaction analysis and MM-GBSA energy calculations are given in the Supplementary Methods section (Supplementary Methods, Results and Discussion).

**FGFR2-binding site residues definition.** Definitions of FGFR2-binding site regions were obtained from the KLIFS database[66].

**Data visualization.** Results were plotted with Seaborn library for Python[67]. Protein structures were visualized with PyMOL (PyMOL Molecular Graphics System, Version 2.5.2 Schrödinger, LLC.) Graphical representations of figures were arranged using Adobe Illustrator©.

### Statistical analysis

Data were analyzed using GraphPad Prism 7 and 9 (GraphPad Software Inc, CA, US) and are shown as mean ± standard deviation (SD). Statistical analysis was performed using unpaired, two-tailed Student's $t$ test and one-way ANOVA as appropriate, unless stated otherwise. $IC_{50}$ values were generated using nonlinear regression analysis (dose-response inhibition) by comparing the inhibitor with a normalized response assuming a variable slope. All experiments were independently performed at least three times, and $P < 0.05$ was accepted for statistical significance.

### Reporting summary

Further information on research design is available in the Nature Portfolio Reporting Summary linked to this article.

## Data availability

The in-silico data generated in this study have been deposited in the Zenodo database (https://zenodo.org/records/7456830) The authors declare that all data supporting the findings of this study are available within the Article, Supplementary Information, or Source Data file. Source data are provided with this paper. Our ethical approval does not allow the complete upload of the results from patient DNA sequencing. All relevant information from the DNA sequencing are included in the manuscript. Source data are provided with this paper.

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

## Acknowledgements

Authors were supported by the German Research Foundation (DFG) grant SFB/TR 209 (M.B, K.S.O. and N.P.M.), Else Kröner-Fresenius Foundation (Else Kröner Forschungskolleg Tubingen (2015_Kolleg_14), S.S. and M.B), German Cancer Aid (70114571, S.S.), the Forum Gesundheitsforschung of the Baden-Württemberg State Ministry of Science, Research and Arts (32-5400/58/2, M.B. and K.S.O.), the Ministry of Baden-Württemberg for Economic Affairs, Labor and Tourism (3-4332.62-HSG/84, A.S. and M.T.) and the German Ministry for Education and Research (eMed, Multiscale HCC (FKZ 01ZX1601G) to M.B. and N.P.M.). We acknowledge support from the Open Access Publication Fund of the University of Tübingen.

## Author contributions

S.S. and M.B. conceived and designed the study. S.S., F.K., Y.R., C.G. and K.C.B. conducted in vitro experiments. S.S. F.K. and M.B. analyzed data and performed statistical analyses. A.S. and M.T. performed and analyzed DigiWest experiments. K.R., M.K., H.S-M. and S.B. conducted patient data and sample collection as well as medical evaluation and analysis. M.H. evaluated the radiologic responses. E.S., T.K., S.A.L. and A.P. performed and analyzed the in-silico modeling. S.S. and M.B. wrote the first draft of the manuscript. E.S., K.S-O., N.P.M and A.P. contributed to data interpretation and manuscript edit. All authors critically reviewed and approved the manuscript.

## Funding

## Competing interests

M.B. reports that he received compensations as a member of scientific advisory boards of Roche Pharma AG, Incyte Biosciences Germany

GmbH, Bayer Vital GmbH, Bristol-Myers Squibb, MSD Sharp & Dome, Taiho oncology Europe, outside the submitted work. The remaining authors declare no potential conflicts of interest.

## Additional information

[1]Department of Internal Medicine I, University Hospital Tuebingen, 72076 Tuebingen, Germany. [2]Department of Pharmaceutical and Medicinal Chemistry, Institute of Pharmaceutical Sciences, Eberhard-Karls-University, 72076 Tuebingen, Germany. [3]Tuebingen Center for Academic Drug Discovery & Development (TüCAD2), 72076 Tuebingen, Germany. [4]NMI Natural and Medical Sciences Institute at the University of Tuebingen, 72770 Reutlingen, Germany. [5]Center for Personalized Medicine, Eberhard-Karls University, 72076 Tuebingen, Germany. [6]CeGaT GmbH and Praxis für Humangenetik, 72076 Tuebingen, Germany. [7]Cluster of Excellence, Image Guided and Functionally Instructed Tumor Therapies, Eberhard-Karls University, 72076 Tuebingen, Germany. [8]Department of Diagnostic and Interventional Radiology, Eberhard-Karls University, 72076 Tuebingen, Germany. [9]Department of Molecular Medicine, Interfaculty Institute for Biochemistry, Eberhard-Karls University, 72076 Tuebingen, Germany. [10]German Cancer Consortium (DKTK) and German Cancer Research Center (DKFZ), 69120 Heidelberg, Germany. [11]M3-Research Center for Malignome, Metabolome and Microbiome, Eberhard-Karls University, 72076 Tuebingen, Germany. [12]School of Pharmacy, University of Eastern Finland, 70210 Kuopio, Finland. [13]These authors contributed equally: Stephan Spahn, Fabian Kleinhenz. ✉e-mail: stephan.spahn@med.uni-tuebingen.de; michael.bitzer@uni-tuebingen.de

