## [Peer Review File · Nature Communications]

THE MOLECULAR INTERACTION PATTERN OF LENVATINIB ENABLES INHIBITION OF WILD-TYPE OR KINASE-MUTATED FGFR2-DRIVEN CHOLANGIOCARCINOMAREVIEWER COMMENTS

Reviewer #1 (Remarks to the Author):

The authors describe the use of Lenvatinib as FGFR2 inhibitor with activity against mutants related to cholangiocarcinoma.

As a medicinal computational chemist, I will comment on this part of the work only. The modeling study is very well done and provide useful information when comparing Lenvatinib to other inhibitors. I have two minor comments:

1. There is no chemical structure given. In its current state, the medicinal chemistry is therefore a little hard to follow. For example, what are the differences between Lenvatinib and Pemigatinib ?
2. As another minor comment, embedding the figures in the text would have made it a lot easier to read and review. I would recommend the author to do it in the future.

Reviewer #2 (Remarks to the Author):

Spahn and colleagues observed several tumor responses to lenvatinib, a multi-tyrosine kinase inhibitor, in a small group of patients with FGFR2 altered cholangiocarcinoma. They take this observation back to the laboratory and show that this can be validated in cell line models and note that lenvatinib also maintains effectiveness when several resistance mutations to FGFR-specific inhibitors are co-engineered with FGFR2 fusions. They conclude the manuscript with a clinical example for a patient who developed progressive disease while receiving pemigatinib and subsequently experienced a partial response to lenvatinib. Comments for consideration are included below:

In the Figure 2D, what is the relevance of the gemcitabine sensitivity data? The sensitivity appears to be different in the different fusion expressing cell lines. How is this informing sensitivity to FGFR inhibitors and what mechanistically would explain these differences in sensitivity to a chemotherapy agent?

In Figure 3D/E, a number of differences are noted using the "DigiWest" approach. Were any of these validated by an orthogonal method?

On line 240, what does "functional differences" refer to? Would clarify what is meant by this statement for the different cell lines.

In the text, when the authors note that resistance mutations were or were not found in patients, it is important that they note whether this was by tumor biopsy, cfDNA, or both. The use of cfDNA is more likely to detect polyclonal resistance mechanisms and comparing cfDNA and tumor sequencing data across patients is not appropriate.

In Figure 4D/E, it appears that selective TKI and non-selective TKI were each compared with DMSO. Can the selective and non-selective TKIs be compared to one another to determine what is different about the non-selective TKIs, i.e. whether their efficacy is related to differential inhibition of other kinases or due to better inhibition of FGFR2 in the setting of resistance mutations. This seems like a key issue, as it would guide future drug development strategies in these patients.

Are the authors able to comment on how FGFR2 point mutations versus fusions might impact the sensitivities to selective TKIs and non-selective TKIs?

Th authors note that lenvatinib does not lead to the same toxicities as the FGFR1-4-specific inhibitors. What is the hypothesized mechanism by which Lenvatinib does not cause the same toxicities as more FGFR-specific inhibitors. Is there a different spectrum of FGFR inhibition that spares FGFR1, inhibition

of which is generally the cause of hyperphosphatemia? How is this linked or not to the efficacy findings?

What spectrum of resistance mutations to FGFR-specific inhibitors would lenvatinib be hypothesized to overcome? Any that would not be hypothesized to be overcome with this approach? How does this relate to the different resistance mutations that can be seen with pemigatinib/infigratinib compared with the irreversible FGFR1-4 inhibitor, futibatinib? Given multiple FGFR-specific inhibitors are approved for use in FGFR2 fusion-positive cholangiocarcinoma, how would the authors suggest Lenvatinib be incorporated into treatment algorithms given their data? Should further clinical trials be performed and how would the authors suggest these be designed?

Reviewer #3 (Remarks to the Author):

The manuscript by Spahn and colleagues reports on the effect of lenvatinib in FGFR2-driven cholangiocarcinoma (CCA).

Selective tyrosine kinase inhibitors (TKIs) are approved for the treatment of CCA. Here the authors use the non-selective TKI lenvatinib to treat CCA patients and investigate the effect of FGFR2 fusion proteins in mouse NIH/3T3 cells. A comparison of selective and non-selective inhibitors is performed. Gate-keeper mutations are introduced into the fusion proteins and further investigated.

Selective inhibitors were developed to reduce side-effects from off-target inhibition. In that respect, the proposal to use lenvatinib feels a step backwards. If lenvatinib is targeting FGFRs efficiently, the same side effects and secondary mutations observed for selective inhibitors are likely to occur, as well as additional side effects due to its non-selectivity. An advantage, however, as reported in this manuscript, is that lenvatinib's interaction with the kinase domain seems more flexible, and may also inhibit resistance-mutated FGFR2. However, this has already been shown to be the case for the selective inhibitor futibatinib (Goyal et al. 2019). This is actually also indicated in this manuscript (e.g. Fig 4C). It has also been shown that the selective FGFR inhibitor Debio 1347 retains activity against gate-keeper mutations.

The manuscript describes a very small cohort of 7 patients, including the description of one patient that has been published before. An additional patient with resistance mutation was successfully treated with lenvatinib. This was however not a gate-keeper mutation, which seems to be the focus of most of the manuscript (Fig 6).

Major comments:

1. Figure 3A and B. For the selective inhibitors, the lowest concentrations applied reduced cell mass to under 50% already. Even lower concentrations are needed to estimate the correct IC50.
2. It is claimed in the text that pSTAT3 is not inhibited by selective inhibitors (p. 9, line 219). However, in suppl. Figure 4A pSTAT3 shows a very similar response to selective as non-selective.
3. The DigiWest results shown for pFGFR in Figure 3C seem to be significant. However, looking at the data in suppl. Table 1, they are not. Please explain.
4. The P values presented in suppl. Table 1, does not correspond to the significance levels reported in suppl. Fig 4?

Minor comments:

1. Control the Western blot in figure 2b with kinase inhibitors. To control the specificity of phospho-specific-antibodies, samples treated with FGFR kinase inhibitors should be shown.
2. Orthographic error: p11. Line 261, dephosphorylation

Reviewer #4 (Remarks to the Author):

The manuscript is a nice bed to bench manuscript, which includes multi-disciplinary results. FGFR is of high interest, with several biotech companies developing (mutant) selective compounds. Overall the story is well written, but I see two major flaws:

1. There is no biochemical data of FGFR2 mutants to support the in silico analysis, this should either be done, or the in silico analysis should be toned down. MM-GBSA is as far as I know not quantitative enough. Specifically the conclusion: "results suggest that lenvatinib's superior inhibitory performance.." Alternatively the selective

2. The potency/IC50 of lenvatinib is on the low side $>1\mu\text{M}$. So it is more likely that the efficacy of lenvatinib is due to the unselective nature, as also highlighted by the proteomic analysis. This should be discussed more.

Minor:

The MMGBSA- analysis normalized by HAC is misleading, wrt affinity this has no/little meaning. (in the context of lead optimization it would have but that is not done in this manuscript. The MMGBSA scores (energies) should be shown as they are calculated.

Why was TIP3P chosen, if I'm not mistaken OPLS forcefields have been optimized vs SPC. Single simulations (per ligand, mutant, combis) were run, how reproducible are the simulations? (different replicates with seeds?)

"..torsional conformation in rotatable bonds"  call this torsional profile.

Reviewer #1

"The modeling study is very well done and provide useful information when comparing Lenvatinib to other inhibitors. I have two minor comments:"

1. There is no chemical structure given. In its current state, the medicinal chemistry is therefore a little hard to follow. For example, what are the differences between Lenvatinib and Pemigatinib ?

We thank the reviewer for this comment and understand the importance of providing clear and concise information about the ligands studied in our work. Indeed, the 3D chemical structures of the ligands have been presented in the original manuscript in *Figure 5C*. Additionally, the chemical structures of each ligand were shown in the former *Suppl. Fig. 13* for lenvatinib, *Suppl. Fig. 14* for infigratinib, and *Suppl. Fig. 15* for pemigatinib.

In response to the reviewer's concern and to ensure that readers can readily access these crucial details, we have introduced the following reference in the Results section of our manuscript:

"Despite previous indications of its significance in drug resistance, our investigation did not reveal notable differences in the interactions involving the molecular brake across selected TKIs (chemical structures are shown in Suppl. Fig. 13-15) and mutations (Suppl. Tab. 1)." (Line 293, page 12)

2. As another minor comment, embedding the figures in the text would have made it a lot easier to read and review. I would recommend the author to do it in the future."

We appreciate the reviewer's feedback regarding the presentation of figures in the manuscript. We acknowledge that embedding the figures directly within the text would enhance the readability and review process. However, in our case, we encountered certain space constraints within the main manuscript that prompted us to include additional information in the supplementary section of the paper.

With this in mind, we have strived to ensure that our revised manuscript addresses these concerns by supplying all pertinent information in a readily accessible manner. This approach aligns with the publication guidelines of Nature Communications, allowing readers and reviewers to access the supplementary materials for comprehensive insights while maintaining the core content within the main manuscript. We hope that this approach optimally balances the need for both streamlined presentation and comprehensive detail.

Reviewer #2

Comments for consideration are included below:

1. In the Figure 2D, what is the relevance of the gemcitabine sensitivity data? The sensitivity appears to be different in the different fusion expressing cell lines. How is this informing sensitivity to FGFR inhibitors and what mechanistically would explain these differences in sensitivity to a chemotherapy agent?

Thank you for noting that we did not discuss this interesting finding in the manuscript. We did not investigate the effect sufficiently to explain the observed difference in sensitivity. A recent retrospective clinical study by Abou-Alfa et al. described a shorter PFS to standard chemotherapies for FGFR2-fusion positive CCA patients compared to patients without FGFR2 alterations (Abou-Alfa et al., Targeted Oncology 2022).

In line with our observation that Lenvatinib seems to lead to a longer PFS than Gemcitabine/Cisplatin this further supports the need for large randomized trials to compare FGFR2-inhibiting therapies to standard therapy in the first-line setting. These studies are currently underway, for example, NCT04093362.

However, as we can only speculate about the underlying mechanisms of our in vitro observation, we deleted Figure 2D and the short description in the main text to not distract from our main story.

Deleted sentence: Moreover, cell viability assays demonstrated that expression of all three FGFR2 fusion genes significantly reduced the sensitivity to gemcitabine, one of the standard chemotherapeutic agents in CCA (Fig 2D).

2. In Figure 3D/E, a number of differences are noted using the “DigiWest” approach. Were any of these validated by an orthogonal method?

All protein expression data were generated using the DigiWest technique, and additional validation using an orthogonal method was not performed.

The DigiWest is a variation of the classical Western Blot (WB) that uses a different readout system (for details, see Treindl et al., *Nat Commun* 2016;7:12852). This paper includes a detailed comparison of WB and DigiWest data, and a side-to-side comparison for approximately 100 antibodies confirms the comparability of the methods. Therefore, we do not see the necessity to validate the obtained results with classical WB.

Mass spectrometry does present another obvious orthogonal method but was not employed here. In a recent publication (Kling et al.; *Biol Chem* 2021;403:331-343) we showed good comparability of these very different methods. We would also like to note that care was taken to employ only commercially available antibodies (please see Supplementary File) that have been qualified for use in both WB and DigiWest and that a highly standardized assay system was used to generate the protein expression data.

3. On line 240, what does “functional differences” refer to? Would clarify what is meant by this statement for the different cell lines.

Thank you for noting the inexactness of the used term. With “no functional differences” we meant that all cell lines expressing wt or mutant FGFR-AHCYL2 show increased proliferation compared to empty vector control cells and similar expression levels of the (mutated) FGFR2-fusion protein. Please see Supplementary Figure 6.

To clarify our statement, we updated this sentence in the manuscript:

“Cellular proliferation or expression levels of (mutated) FGFR2-fusion proteins were noted to be similar between cells transfected with FGFR2-AHCYL2 plus p.V564 or plus p.E565A or FGFR2-AHCYL2 without these mutations (Suppl. Fig. 6)”. **(Line 244-246, page 10)**

0. In the text, when the authors note that resistance mutations were or were not found in patients, it is important that they note whether this was by tumor biopsy, cfDNA, or both. The use of cfDNA is more likely to detect polyclonal resistance mechanisms and comparing cfDNA and tumor sequencing data across patients is not appropriate.

We agree that describing the origin of the sequenced material is essential. For the patient with the FGFR2-AHCYL2 fusion, for example, we state that a liver biopsy of the progressive lesion under lenvatinib did not detect any resistance mutation (see Figure 1D).

To further clarify this in the main text of the manuscript, we added in the legend of Figure 1A that the detected molecular alterations were from tumor biopsies (page 30, line 845), and we updated the following sentence: "A sequential liver biopsy of the progressive lesion did not find any FGFR2 mutations as a potential explanation for tumor progression." (Line 143-144, page 6).

For the patient with the FGFR2-BICC N549K resistance mutation, we also added a note to the figure 6 legend that the mutation was detected from a liver biopsy ("after progressive disease to pemigatinib in the presence of an N549K kinase mutation detected in a liver biopsy of a progressive lesion." Line 940, page 38). This complements the existing description in the main text of our manuscript: "A further biopsy of a liver lesion was then performed that revealed a FGFR2 N549K brake mutation." (Line 328, page 13)

5. In Figure 4D/E, it appears that selective TKI and non-selective TKI were each compared with DMSO. Can the selective and non-selective TKIs be compared to one another to determine what is different about the non-selective TKIs, i.e. whether their efficacy is related to differential inhibition of other kinases or due to better inhibition of FGFR2 in the setting of resistance mutations. This seems like a key issue, as it would guide future drug development strategies in these patients.

This is a very important and valid point; we thank the reviewer for this comment. We have addressed this remark as follows:

An additional analysis, a direct comparison of Lenvatinib and Infigratinib, was performed based on the data already presented in Fig. 4D/E. This new analysis, **which is now added as a new Figure 4F**, revealed that phospho-FGFR, phospho-mTOR, mTOR, and phospho-eIF4E were significantly inhibited by lenvatinib when compared to infigratinib (see new Fig 4F, Supplementary File). This underscores the differential effect on FGFR2 phosphorylation and highlights the persistent inhibition of downstream pathways of lenvatinib in the presence of p.V564F resistance mutations. Looking for example on mTOR, lenvatinib is no direct inhibitor, therefore this finding, shown in in the **new Fig. 4F**, supports our hypothesis that lenvatinib's sustained activity in the setting of resistance mutations seems to be mainly due to conserved inhibition of FGFR2 itself.

New Figure 4F:

New Fig. 4F: Volcano Plot and hierarchical cluster (HCL) analysis of proteins and phosphoproteins that significantly differed upon direct comparison of lenvatinib (n=4) and infigratinib (n=4)-treated p.V564F FGFR2-AHCYL2 samples (T-test, Welch, $P \leq 0.05$). For details see new Suppl. Table 3c. Expression values were normalized to total protein signal, median-centered and Log-2 transformed. HCL was performed using Pearson correlation and complete linkage.

We mention this finding in the main text of the manuscript on **page 11, lines 270-76**):

[...] which are mainly involved in PI3K/AKT/mTOR signaling [...]. This was underlined by directly comparing the two treatments to each other. Lenvatinib exhibited a conserved inhibitory differential effect for phosphorylation of FGFR2 (Tyr653/Tyr654), mTOR (Ser2481), eIF4E (Ser209) as well as total mTOR (Fig 4F, Supplementary File). This not only underscores the differential effect on FGFR2 phosphorylation (see Fig.4C) but also highlights the persistent inhibition of downstream pathways of lenvatinib, suggesting that the sustained activity of lenvatinib is likely due to conserved direct FGFR2 inhibition even in the presence of the p.V564F mutation.

6. Are the authors able to comment on how FGFR2 point mutations versus fusions might impact the sensitivities to selective TKIs and non-selective TKIs?

We thank the reviewer for this important question. Several mechanisms, like activating point mutations outside the kinase domain of FGFR2 or extracellular domain in-frame deletions (Cleary et al., Cancer Discov 2021), have been reported as alternative mechanisms of FGFR2 activation in tumor cells. Two of our lenvatinib-treated patients had cholangiocarcinoma with activating point mutations outside the kinase domain (p.C382R and p.S372C), and one tumor harbored an extracellular domain in-frame deletion (370_371delinsCys & DEL). As described in the manuscript, all three patients responded to lenvatinib (see Fig. 1A and 1B, PET-CT scan of the patient with the p.S372C mutation). However, we assume the patient's FGFR2-driven tumors would have also responded to FGFR-selective TKIs.

As our work focuses on the development of therapy resistance during the treatment with FGFR-inhibiting TKIs, we did not experimentally compare the efficacy of selective and non-selective TKIs on activating mutations outside the kinase region or in the context of domain-in-frame deletions, which we see far beyond the scope of this work. Therefore, we cannot comment on this issue in more detail.

7. The authors note that lenvatinib does not lead to the same toxicities as the FGFR1-4-specific inhibitors. What is the hypothesized mechanism by which Lenvatinib does not cause the same toxicities as more FGFR-specific inhibitors. Is there a different spectrum of FGFR inhibition that spares FGFR1, inhibition of which is generally the cause of hyperphosphatemia? How is this linked or not to the efficacy findings?

We thank the reviewer for this clinically relevant and interesting question. As the reviewer already assumes, a different profile of FGFR1-4 inhibition exists between the investigated drugs. To highlight this aspect, we show the information on the IC50 values for FGFR1-4 inhibition for each substance from selected publications in a new supplementary table (Suppl. Tab. 4). It is well known that hyperphosphatemia results from FGFR1 inhibition in the context of pemigatinib or infigratinib. When looking at the IC50 profiles, it is apparent that lenvatinib has a higher inhibition efficiency of FGFR2 compared to FGFR1. For hyperphosphatemia, which is in contrast to pemigatinib or infigratinib, we hypothesize that this is most likely one explanation for why lenvatinib causes different toxicities.

As this is an important insight, we added the following sentence to the discussion: “In contrast to the selective FGFR inhibitors infigratinib and pemigatinib, lenvatinib has a higher inhibition efficiency of FGFR2 than FGFR1 (see Suppl. Table 4). We hypothesize that this could be one reason for the different spectrum of adverse effects”. (Line 415-418, page 18)

Information in new Supplementary Table 4:

	IC ₅₀ pemigatinib	IC ₅₀ infigratinib	IC ₅₀ lenvatinib
FGFR1	0.4nM	0.9nM	61nM
FGFR2	0.5nM	1.4nM	27nM
FGFR3	1nM	1nM	52nM
FGFR4	30nM	60nM	43nM
Ratio FGFR2/FGFR1	1.3	1.6	0.4

4. What spectrum of resistance mutations to FGFR-specific inhibitors would lenvatinib be hypothesized to overcome? Any that would not be hypothesized to be overcome with this approach? How does this relate to the different resistance mutations that can be seen with pemigatinib/infigratinib compared with the irreversible FGFR1-4 inhibitor, futibatinib? Given multiple FGFR-specific inhibitors are approved for use in FGFR2 fusion-positive cholangiocarcinoma, how would the authors suggest Lenvatinib be incorporated into treatment algorithms given their data? Should further clinical trials be performed and how would the authors suggest these be designed?

We thank the reviewer for these far-reaching questions. To expand our data, we explored four additional mutations mentioned in a recent publication on Futibatinib by Goyal et al. (N Engl J Med 2023;388:228-39): N549D, V562L, V564I, and E565G.

We now show the results of these additional mutations in Suppl. Tab. 1B and 2B, and extended the Suppl. Fig. 7-15 with this new information. Of note, these additional investigations revealed similar results for all these mutations. These observations imply that lenvatinib might have clinical activity in FGFR2-driven tumors despite the presence of all investigated mutations (N549K, N549D, E565A, E565G, V562L, V564F, and V564I). One example of this hypothesis is the described patient in the manuscript with progression during the treatment with pemigatinib due to a FGFR2-BICC N549K resistance mutation who responded to lenvatinib as a second-line FGFR2 inhibition.

Our observation stimulates several crucial new research questions that have not been investigated by clinical trials so far.

First, the increased flexibility of lenvatinib could lead to delayed development of secondary resistances. An additional hypothesis might be that in contrast to the selective FGFR inhibitors, resistance to lenvatinib might be primarily caused by alternative mechanisms than FGFR2 kinase mutations. In HCC, for example, lenvatinib resistance has been linked to activating the EGFR-PAK2-ERK5 signaling axis (Jin et al., Nature 2021; 595:730-34).

This interesting hypothesis of delayed development of FGFR2 resistance mutations during the treatment with lenvatinib compared to pemigatinib or infigratinib could be investigated in an exploratory clinical trial with repeated liquid biopsies during treatment.

Second, in the case of progression on pemigatinib or infigratinib due to resistance mediating kinase mutations, lenvatinib could be compared to futibatinib, a selective covalent FGFR-inhibitor with known activity against a spectrum of infigratinib or pemigatinib resistance mediating mutations (Goyal et al., N Engl J Med 2023;388:228-39). A primary study objective in that setting could be the treatment response to either substance.

As these studies would reveal important information that is necessary to define an ideal inclusion of lenvatinib to the treatment algorithm in FGFR2-driven CCA, we included the following sentence in the discussion part of the manuscript:

“Our results stimulate new hypotheses for exploratory studies that could guide the optimal inclusion of lenvatinib in the treatment algorithm of FGFR2-driven CCA, such as to compare the appearance of kinase resistance mutations during the treatment with lenvatinib or specific-FGFR inhibiting TKIs by repeated liquid biopsies or therapy responses despite the presence of resistance-mediating mutations.” (Line 419-423, page 18)

Reviewer #3:

Selective inhibitors were developed to reduce side-effects from off-target inhibition. In that respect, the proposal to use lenvatinib feels a step backwards. If lenvatinib is targeting FGFRs efficiently, the same side effects and secondary mutations observed for selective inhibitors are likely to occur, as well as additional side effects due to its none-selectivity. An advantage, however, as reported in this manuscript, is that lenvatinib’s interaction with the kinase domain seems more flexible, and may also

inhibit resistance-mutated FGFR2. However, this has already been shown to be the case for the selective inhibitor futibatinib (Goyal et al. 2019). This is actually also indicated in this manuscript (e.g. Fig 4C). It has also been shown that the selective FGFR inhibitor Debio 1347 retains activity against gate-keeper mutations.

The manuscript describes a very small cohort of 7 patients, including the description of one patient that has been published before. An additional patient with resistance mutation was successfully treated with lenvatinib. This was however not a gate-keeper mutation, which seems to be the focus of most of the manuscript (Fig 6).

Major comments:

1. **Figure 3A and B. For the selective inhibitors, the lowest concentrations applied reduced cell mass to under 50% already. Even lower concentrations are needed to estimate the correct IC50.**

We agree that for a more precise determination of the IC50 values, lower concentrations of the selective inhibitors should have been tested. We, therefore, repeated the experiment for infigratinib with lower concentrations and recalculated the IC50 value.

When we, for example, added concentrations of 0.01, 0.05, and 0.1 μM to the already used 0.25, 0.5, 1, and 5 μM concentrations of infigratinib, the estimated IC50 values were nearly identical with 133 nM in this new experiment, compared to 133nM in Figure 3.

As these minimal (<1nM) differences have no impact on the reported results and conclusions, in our eyes, it is best to keep the calculated values of the original submitted version, as all experiments for the reported IC50 calculations were done under identical conditions and are thus in our view more robust and better controlled. However, we provide, for review only, the results of the additional experiment below with the additional concentrations of 0.01, 0.05, and 0.1 μM (red line) compared to the results from Figure 3 (black line).

- 2. It is claimed in the text that pSTAT3 is not inhibited by selective inhibitors (p. 9, line 219). However, in suppl. Figure 4A pSTAT3 show a very similar response to selective as non-selective.**

We thank the reviewer for this valuable comment and have carefully looked at this point.

The statement in the text was based on the DigiWest results depicted in Figure 3D and E. Here, we only saw a trend towards pSTAT3 inhibition for selective inhibitors ($p=0.0965$, $FC=-1.14$), whereas the expression levels of pSTAT3 for non-selective TKIs were statistically significant ($p=0.0151$, $FC=-1.37$).

We clarify the different statistics applied in more detail in the response to comment 4, but when looking at absolute values and using Welch's ANOVA with Dunnett's Multiple Comparisons, as done for the old Supplementary Fig. 4A (new Supplementary Fig. 5A): STAT3 phosphorylation is inhibited by both, selective ($p=0.025$) and unselective TKIs ($p=0.025$).

We hence agree that we overinterpreted the results for pSTAT3 and prefer to remove this **specific claim and have edited the text accordingly:**

"In contrast, the non-selective TKIs additionally inhibited also MAPK-unrelated proteins such as STAT3 (p-Ser727), p70S6 kinase (Thr389) and S6 ribosomal protein (Ser235/236), which are involved in Jak/STAT or PI3K/AKT/mTOR pathways[...]" **(line 223, page 9)**

- 3. The DigiWest results shown for pFGFR in Figure 3C seem to be significant. However, looking at the data in suppl. Table 1, they are not. Please explain.**

Indeed, we detected a significant reduction of pFGFR (Y653/654) in F-AHCYL2 cells treated with the four different FGFR inhibitors when compared with DMSO. This is shown in Figure 3C. For this analysis we used One-Way-ANOVA as clarified in the corresponding Figure legend (line 888-889, page 34).

In contrast, old supplementary Table 1 (new Supplementary File: DigiWest Fig 3D+E) shows the calculated p-values for the volcano-plots in Figures 3D+E. For these graphs, the compounds used were assigned to different TKI classes. For this analysis, we formed the ratios to the untreated controls (DMSO) and calculated p-values using Welch's T-Test for these groupwise comparisons. The determined p-values are listed in the old Suppl. Table 1. These results have not been used in Figure 3C.

Likely, due to having pooled treatments close to the detection limit (<100 AFI), the statistics applied and shown in sup. Table 1 only detected a trend towards inhibition of pFGFR, however it did not reach statistical significance (Please see Figure below). For clarification, also see the Suppl. Fig. 5A.

Suppl.Fig.5A for pFGFR:

4. The P values presented in suppl. Table 1, does not correspond to the significance levels reported in suppl. Fig 4?

There is indeed a discrepancy in calculated p-values shown in Suppl. Table 1 (based on Fig.3D/E; now part of the Supplementary File: DigiWest Fig 3D+E) and the values used in Suppl. Fig. 5A (former 4A). This is due to the data being analyzed slightly differently.

For Fig3D/E, replicates were compared as a group (Welch's T-Test) using normalized absolute values. For Suppl. Fig. 4A (now 5A), treated samples were set in relation (%) to the respective control (DMSO) sample from the same experiment (sample set), thus correcting for cell culture-induced experimental variation; then groups were compared to DMSO using Welch's ANOVA.

To keep the presented data consistent, we now have updated Suppl. Fig. 5A and included additional tables in the new Supplementary File (DigiWest Sup. Figure 5A) now showing the significance values from these comparisons for key downstream phosphorylation events. The same applies for the table DigiWest 4D+E in the new Supplementary File (old sup. Table 2) (based on Fig. 4D/E) and Suppl. Fig. 5B (Supplementary File: DigiWest Sup. Fig 5B), respectively.

Overall, the data analysis that was applied for the main figure can be regarded as more stringent and robust and the supplementary data shows similar trends.

For better clarification, we merged all supplementary tables depicting our DigiWest results to one new Supplementary File with labeling that clarifies the corresponding Figures.

Additionally, we have clarified this point in the method section :

Page21, Line 501-504: [...] using an Excel-based analysis tool. MEV 4.9.0 was used for heatmap generation and respective statistics (Welch's T-Test, group comparison). For

DigiWest supplementary figures 4A/B, after correction for experimental variation, groups were compared to DMSO controls using Welch's ANOVA with Dunnett's Multiple Comparisons Test.

Page 23, Line 556: [...] using unpaired, two-tailed Student's t-test and One-Way-ANOVA as appropriate, unless stated otherwise. [...] MEV 4.9.0 was used for DigiWest heatmaps⁶⁹.

Minor comments:

- 1. Control the Western blot in figure 2b with kinase inhibitors. To control the specificity of phospho-specific-antibodies, samples treated with FGFR kinase inhibitors should be shown.**

We agree and tested and confirmed the specificity of the phospho-specific antibodies for pFGFR2, pMAPK2 and pSTAT3 through performing Western Blot analysis.

We included the results in the manuscript as a new Supplementary Figure 2 and added the sentence: Phosphorylation of pFGF, ERK1/2 and STAT3 could be reversed through treatment inhibition with infigratinib, confirming the specificity of the phospho-specific-antibodies (Suppl. Fig. 2) to the main text (**page 7, line 178-180**).

Supplementary Figure 2. Western blot analysis of the FGFR2 pathway in FGFR2-AHCYL2 transfected NIH/3T3 cells after 24h treatment with 0.5µM infigratinib (A). Densitometric analyses (B) of pFGFR, pMAPK and pSTAT3 in FGFR2-AHCYL2-NIH/3T3 cell line after treatment with infigratinib, graphs show mean ± SEM, **P ≤ 0.01, ****P ≤ 0.0001 compared to DMSO treated cells. Densitometric signal from DMSO bands were normalized to 100% for each repetition. n = 3.

- 4. Orthographic error: p11. Line 261, d ephosphorylation** We thank the reviewer for noting that orthographic error. We corrected it.

Reviewer #4

Overall the story is well written, but I see two major flaws:

- 1. There is no biochemical data of FGFR2 mutants to support the in silico analysis, this should either be done, or the in silico analysis should be toned down. MM-GBSA is as far as I know not quantitative enough. Specifically the conclusion: "results suggest that lenvatinib's superior inhibitory performance.." Alternatively the selective**

We agree that additional biochemical data, beyond our already presented experimental work, would further strengthen our findings. However, we would like to explain our choice of MM-GBSA calculations for the analysis.

Firstly, we would like to underline that we utilized MM-GBSA calculations not on docking poses but along the MD trajectory. The process is discussed in **Supplementary Methods Results & Discussion (pages 3-4, lines 104-110)**: "Each 5th frame of MD was used for MM-GBSA calculations (402 complexes proceeded for an individual complex x 3 ligands x 8 FGFR2 systems)". Therefore, we have generated energy properties for 9648 individual complexes, which provides a more comprehensive estimate than a single conformation. Moreover, to underline the validity of our calculations, we presented not the single energy data points but the statistically derived MM-GBSA data in the form of box plots (**Suppl. Figures 7-12**). Additionally, a Zenodo repository is included in the original manuscript, where the raw MM-GBSA data can be found, including the per-frame energy properties, as well as the energy components that were not discussed in our work (**page 23, lines 563-566**): "*Zenodo repository (DOI: 10.5281/zenodo.7456830). The available data includes the raw molecular dynamic trajectories and full-component MM/GBSA tables*"

The further discussion of energy components of binding free energy and the connection between them is anticipated in **Supplementary Methods Results & Discussion**, section "*MM-GBSA energy proposes two potential binding lenvatinib conformations in V564F FGFR2*" (**page 6, lines 154-167**).

Although MM-GBSA has limitations, it is a widely used and accepted method for predicting binding affinities, particularly when supported by multiple simulation frames. We are aware that FEP+ would be a superior method; however, it requires precise starting conditions, typically obtained from X-ray crystal structures. Additionally, FEP+ involves performing chemical transitions for ligands. In the context of clinical pharmacology, these requirements pose challenges due to the lack of structural similarity between ligands and the unavailability of appropriate X-ray data for individual mutations. Consequently, methods like MM-GBSA and related approaches offer a more feasible and efficient alternative.

Concerning the remark of the reviewer on lenvatinib's superior inhibitory performance, we agree that this conclusion would have a stronger foundation with further validation through biochemical experiments. However, we would like to emphasize that our in silico analyses provide a strong foundation for understanding the molecular interactions between lenvatinib and FGFR2 mutants in this particular case. In addition, our in silico research is supported by

our biochemical results in the p.V564F FGFR2 cell line. In contrast to infigratinib, lenvatinib demonstrated a sustained inhibition of cell growth (see Fig. 4A) and the sustained inhibition of pFGFR (see Fig. 4C and new Fig 4F) in the presence of the resistance mutation.

To further strengthen our in silico analysis, we extended our Molecular Dynamics (MD) simulations for four further FGFR2 mutants (N549D, V562L, V564I, and E565G) that have been described in the context of therapy resistance or gain of function. The additional simulations with Lenvatinib, Infigratinib, and Pemigatinib led to a similar conclusion on the higher inhibitory performance of lenvatinib.

Finally, we have carefully rewritten and streamlined the in silico modeling part within the results section, including the four additional resistance mutations and further explanations, as we agree with the reviewer that more explanations were needed to follow our argumentations.

2. The potency/IC50 of lenvatinib is on the low side >1 μ M. So it is more likely that the efficacy of lenvatinib is due to the unselective nature, as also highlighted by the proteomic analysis. This should be discussed more.

We thank the reviewer for mentioning this important point. Looking at our proteomic analysis in the wildtype FGFR2-fusion cell line, it is interesting to note that at TKI concentrations that led to similar inhibition of cell growth, the pFGFR inhibition is comparable between lenvatinib and the selective FGFR inhibitors (see Figure 3C). Moreover, thanks to the suggestion of reviewer 2, we now included a direct comparison of our proteomic analysis of infigratinib vs. lenvatinib in p.V564F harboring FGFR2 cell lines in the manuscript (see new Figure 4F). This comparison demonstrates that even in the presence of the resistance mutation p.V564F, lenvatinib significantly inhibits pFGFR, suggesting that the effect of lenvatinib seems to be mainly mediated via FGFR2 in the presence of the gate-keeper mutation.

In our eyes, the proteomic data supports the conclusion that the main effect of lenvatinib in our FGFR2-fusion-bearing cell lines seems to be mediated through FGFR2. But of course, we also detected inhibition of non-typical FGFR2 downstream signaling pathways and agree with the reviewer that this should be discussed more. In our view, this more unselective nature might be beneficial to overcome or even prolong the development of FGFR2 resistance mutations.

We therefore included the point in our discussion of the proteomic results:

Moreover, lenvatinib conferred inhibition of downstream targets in protein profiling analyses, besides typical FGFR2 downstream signaling pathways, which might be important in the context of treatment responses or acquisition of resistance and might be a beneficial additive consequence of the more unselective nature of lenvatinib. (Page 17, lines 395-398)

Minor:

1. The MMGBSA- analysis normalized by HAC is misleading, wrt affinity this has no/little meaning. (in the context of lead optimization it would have but that is not done in this manuscript. The MMGBSA scores (energies) should be shown as they are calculated.

We appreciate the reviewer's comment on the usage of HAC in our MM-GBSA calculations. A critical aspect of MM-GBSA calculations is the consideration of the size of the ligand molecule, which can influence the energy contribution. HAC (Heavy Atom Count) measures the number of non-hydrogen atoms in the ligand molecule. It is often used as a normalization factor in MM-GBSA calculations to account for ligands with different sizes. The choice of HAC in comparing the energy profiles of lenvatinib (mM= 426.9 g/mol), pemigatinib (mM = 487.5 g/mol), and infigratinib (mM=560.5 g/mol) was based on this reason.

Normalizing the energy values by HAC allows for a more meaningful comparison of the relative binding affinities of ligands, regardless of their size, by considering the number of heavy atoms, which are the atoms that contribute most to the ligand-protein interactions. This approach assumes that the ligand-protein interactions scale linearly with the number of heavy atoms in the ligand, which has been shown to hold reasonably well for small ligands.

While we acknowledge the limitations of HAC normalization, we have included other calculations in the original manuscript to support our findings and ensure that our results are not solely based on MM-GBSA analysis. Namely,

- in-depth ligand interaction analysis (**Figure 5A+B, Supplementary Table 2**),
- separate calculations for hydrophobic interaction frequencies (**Supplementary Table 3**), and
- ligand torsional profiles (**Supplementary Figures 13–15**).

To understand the changes, that might have occurred in FGFR2 upon binding different ligands we investigated the key FGFR2 residues, commonly referred as Molecular Brake (**Supplementary Table 1, Supplementary Discussion p. 6-7, lines 181-191**). Additionally, we studied another well-established dynamic element among protein kinases – hydrophobic regulatory R-spine in relation to the lenvatinib, pemigatinib, and infigratinib binding (**Supplementary Table 3A,3B Supplementary Discussion p. 7, lines 193-206**). Notably, all these calculations were performed on a whole Molecular Dynamics trajectory, comprising 2000 complexes per ligand/per system, rather than on a single FGFR2-ligand docking pose.

Furthermore, we have performed additional calculations of the MM-GBSA ΔG Binding Free Energy, which are now included in the revised version of the manuscript (new **Supplementary Figure 8**). Here, no HAC normalization was used. When comparing the ΔG Binding Free Energy values to the original ligand efficacy values, we observed that although the absolute values may vary across specific systems, the ratios of the values between Lenvatinib/Pemigatinib and Infigratinib remain consistent across different systems. This consistency in the distribution trend is noteworthy. The absolute difference in values can be attributed using different scales (ΔG in kcal/mol and ligand efficacy in kcal/mol per heavy atom count) and the disregard of molecular sizes. By including these additional calculations, we aim to emphasize the reliability of our initial energy predictions.

In response to the reviewers' previous comment, we have mentioned that we provide a Zenodo repository which is referenced in the original manuscript. This repository contains the full MM-GBSA calculation output, including the per-frame energy properties and the prime energy components that were not discussed in our work.

We hope these clarifications and recourses will address the reviewer's concerns regarding the MM-GBSA analysis.

2. Why was TIP3P chosen, if I'm not mistaken OPLS forcefields have been optimized vs SPC. Single simulations (per ligand, mutant, combis) were run, how reproducible are the simulations? (different replicates with seeds?)

Thank you for your questions regarding our choice of water model and simulation reproducibility. We chose to use the TIP3P water model in our study based on previous literature experience of the application and validation of the TIP3P water model in protein-ligand simulations with the OPLS force fields (Kaminsiki et al., J Phys Chem 2001;105:6474-6487; Harder et al., PNAS 2008;105:7906-7911; Roe et al., J Chem Theory Comput 2013;9:3084-3095; Lemkul et al., J Physical Chem 2010;114:13309-13318). Additionally, the experience of our research group demonstrated that the combination of the TIP3P water model and the OPLS force fields can accurately describe the behavior of protein-ligand systems, providing reliable insights into their interactions and dynamics (Pantsar et al, Nat Commun 2022;13:569; Pantsar, Sci Rep 2020;10:11992; Goncalves et al., J Chem Inf Model 2022;62:5746-5761).

We agree that other viable options exist, such as SPC, and, for instance, OPC3. We acknowledge that there may be slight differences in the capabilities of these water models to simulate properties of clean water. Still, these differences are minor when simulating complex systems such as protein-ligand interactions.

Regarding force field optimization, the OPLS force field is not optimized for the TIP3P water model. However, OPLS force field is not specifically optimized for SPC either. Instead, the recommended force field parameters for the ligand and protein are chosen, which have been shown to work well with TIP3P water in previous studies. For instance, according to Harder et al. (J Chem Theory Comput 2016;12:281-296), in which the authors note that the force field is suitable for a variety of simulations, including protein-ligand binding, protein folding, and drug discovery studies.

Regarding simulation reproducibility, we performed multiple independent simulations for each ligand, mutant, and combination using different initial seeds. Each individual complex (FGFR2-selected mutant-selected ligand) was simulated for 2ps, resulting in a total simulation time of 24ps (3 ligands x 4 FGFR2 systems x 2ps). In the revised version of the manuscript, we extended the MD simulations to include N549D, V562L, V564I, and E565G FGFR mutants with all three ligands. In summary, for each ligand, we simulated 8 systems for 2ps, resulting in a total simulation time of 48ps (3 ligands x 8 FGFR2 systems x 2ps). The new MD simulations were conducted using the same settings and procedure as described in the original manuscript.

3. "..torsional conformation in rotatable bonds"  call this torsional profile.

Thank you for your suggestion to simplify the description of our analysis. We have updated the terminology from 'torsional conformation in rotatable bonds' to 'torsional profile', as per your recommendation. This change can be found on **pages 12, line 286** and supplementary Results and Discussion **page 5, line 123, as well as in the figure legend of supplementary figure 13-15**

REVIEWERS' COMMENTS

Reviewer #2 (Remarks to the Author):

Thank you for detailed answers to my comments on the initial manuscript submission.

Reviewer #3 (Remarks to the Author):

I thank the authors for answering all of my questions. I have no further comments.

Reviewer #4 (Remarks to the Author):

The authors have addressed the comments from my previous review, and I would recommend the article for further consideration to publish, nice work.